# Network control energy reductions under DMT relate to serotonin receptors, signal diversity, and subjective experience

S. Parker Singleton [1] ✉, Christopher Timmermann [2], Andrea I. Luppi [3], Emma Eckernäs [4], Leor Roseman[2], Robin L. Carhart-Harris[2,5] & Amy Kuceyeski [1,6]

Psychedelics offer a profound window into the human brain through their robust effects on perception, subjective experience, and brain activity patterns. The serotonergic psychedelic N,N-dimethyltryptamine (DMT) induces a profoundly immersive altered state of consciousness lasting under 20 min, allowing the entire experience to be captured during a single functional magnetic resonance imaging (fMRI) scan. Using network control theory, we map energy trajectories of 14 individuals undergoing fMRI during DMT and placebo. We find that global control energy is reduced after DMT injection compared to placebo. Longitudinal trajectories of global control energy correlate with longitudinal trajectories of electroencephalography (EEG) signal diversity (a measure of entropy) and subjective drug intensity ratings. At the regional level, spatial patterns of DMT's effects on these metrics correlate with serotonin 2a receptor density from positron emission tomography (PET) data. Using receptor distribution and pharmacokinetic information, we recapitulate DMT's effects on global control energy trajectories, demonstrating control models can predict pharmacological effects on brain dynamics.

Serotonergic psychedelics such as lysergic acid diethylamide (LSD), psilocybin, and N,N-dimethyltrypamine (DMT) are powerful neuromodulators that transiently alter human experience[1,2] and have shown potential for treating a variety of common affective and addictive disorders[3,4]. DMT is a naturally occurring tryptamine and is the primary psychoactive compound found in ayahuasca, a ceremonial brew used for hundreds of years in South America[5]. Unlike LSD and psilocybin, DMT is rapidly metabolized in the body by monoamine oxidase (MAO) enzymes, requiring it to be combined with MAO inhibitors in order to be orally active, as is the case in ayahuasca. This results in a DMT experience that rises and falls over the course of several hours, similar to oral LSD and psilocybin. When inhaled or injected intravenously at large enough doses, however, DMT rapidly produces immersive "breakthrough" experiences characterized by vivid and complex visual imagery - occurring within one minute of administration - and lasting for only 15–30 min[6–9]. This provides a unique opportunity to study human brain dynamics during the onset, peak, and offset of DMT's effects over a single functional scan.

Human neuroimaging studies with LSD and psilocybin have demonstrated that the psychedelic state is one of prominent reorganization of brain dynamics[10–13]. These compounds acutely decrease functional connectivity within the brain's functional sub-networks, while increasing functional connectivity between functional sub-networks[13–17]. The impact of psychedelics on subjective experience[18], neural dynamics[19,20], and therapeutic behavior change[21] has been linked to agonism of the serotonin 2a (5-HT2a) receptor. This finding affords a unique opportunity to model and study the perturbation of brain dynamics using whole-brain computational models that incorporate receptor distribution information. Such whole-brain models have highlighted the central role of the spatial distribution of the 5-HT2a receptor in the shift in brain dynamics under LSD and psilocybin[22–24].

Network control theory is a linear dynamical systems approach that models state transitions occurring within a network[25]. When applied to the brain, typically the structural connectivity matrix derived from diffusion MRI (dMRI) is the network over which transitions between functional brain states are modeled. These functional brain states may be theoretical, e.g., activations of a priori functional networks[26,27], or empirical, e.g., statistical brain maps derived from task functional MRI (fMRI)[28,29] or commonly recurring patterns of co-activation derived from the clustering of task-free

[1]Department of Computational Biology, Cornell University, Ithaca, NY, USA. [2]Center for Psychedelic Research, Department of Brain Science, Imperial College London, London, UK. [3]Montreal Neurological Institute, Montreal, ON, Canada. [4]Unit for Pharmacokinetics and Drug Metabolism, Department of Pharmacology, Sahlgrenska Academy at University of Gothenburg, Gothenburg, Sweden. [5]Psychedelics Division, Neuroscape, University of California San Francisco, San Francisco, CA, USA. [6]Department of Radiology, Weill Cornell Medicine, New York, NY, USA. ✉e-mail: sps253@cornell.edu

fMRI time-series[24,30]. Control energy is defined as the amount of input needed to drive the system from one state to another. Overall, network control theory has demonstrated utility in describing brain dynamics in a variety of cognitive states[30,31] and neuropsychiatric/degenerative conditions[26,28,32,33], as well as throughout development[27,34] and during neuromodulation and pharmacologically induced altered states[24,29,35].

Receptor-informed network control theory is an extension we recently deployed in order to model the effects of LSD and psilocybin on brain activity dynamics. We found that the acute administration of LSD and psilocybin reduces the control energy required to transition between task-free fMRI-derived brain-states in a manner that, across individuals, covaries with increases in brain activity entropy - i.e., the diversity or complexity of the brain's spontaneous oscillations recorded across time, a well-known marker of psychedelic action[36]. Reduced control energy here indicates more facile state transitions under the network control framework, thereby potentially enabling access to more complex sequences of brain activity. Moreover, we provided evidence that the reduced control energy effect of psychedelics is associated with the brain's spatial distribution of the 5-HT2a receptor expression[24]. In that work, we studied the transitions between and dynamics of four representative activity patterns. While this approach is ideal for summarizing overall changes, it lacks the temporal resolution necessary for capturing instantaneous, moment-to-moment shifts in dynamics under a rapidly changing cognitive state, such as when under the influence of DMT.

In the present analysis, our aims are twofold. First, we seek to quantify DMT's effects on control energy and their relation to the serotonin 2a receptor, neural entropy, and subjective effects. Given the rapid kinetics of DMT's effects, the use of time-resolved analysis techniques will be crucial for capturing changes in the brain's activation dynamics in real time. Therefore, our second aim is to evolve our methods in order to address this challenge. Here, we employ a time-resolved network control analysis of $N = 14$ healthy individuals undergoing 28 min of simultaneous electroencephalography (EEG) and fMRI recordings for 8 min before and 20 min after an intravenous (I.V.) bolus injection of DMT and (on a separate visit) placebo (Fig. 1a)[37]. These multimodal and continuous scanning conditions enable high temporal (EEG) and spatial (fMRI) resolution of brain activity before, during, and after an injection of DMT. Herein, we expand upon our prior work with LSD and psilocybin[24] by deploying a time-resolved network control analysis of the entire trajectory of the effects of DMT (Fig. 1b). We compare control energy dynamics between DMT and placebo, observe temporal trajectories of these dynamics, and relate these changes to contemporaneous changes in neuronal signal diversity (Lempel-Ziv complexity; a measure approximating entropy) from concurrently-acquired EEG. Further, we compare DMT-related changes in regional dynamics to various serotonin receptor maps, including 2a. Lastly, we demonstrate an ability to simulate the impacts of DMT on control energy dynamics in silico using only fMRI data from placebo scans, 2a receptor density information, and pharmacokinetic modeling of DMT plasma concentrations.

## Results

We analyzed simultaneous EEG-fMRI resting-state data for 14 participants acquired across two visits (one drug visit and one placebo visit), each on separate days[37]. At each visit, two scanning sessions comprising 28-min-long resting-state EEG-fMRI scans were collected, with I.V. bolus infusion of either DMT or placebo at the end of the 8th minute (Fig. 1a). The second scanning session included the collection of subjective drug intensity ratings (0–10) at the end of every minute, while the first scanning session was resting-state. This report analyses the EEG-fMRI data of the resting-state scan while using the intensity ratings of the second session for correlational analyses. See "Participants and study design" and "EEG-fMRI acquisition" sections for full details on study design and acquisition.

### Global control energy is lower after DMT infusion versus after placebo

We first begin by computing a control energy time-series from each participant's 28 min resting-state DMT and placebo fMRI scans. Control energy here is defined as the amount of input needed to drive the system from the current brain activity pattern to the next, where each brain activity pattern is a regional vector summarizing a single brain volume within the fMRI (Fig. 1b; see "Control energy calculation" section for details). We find that DMT control energies are significantly lower than placebo control energies for a majority of the time-points (61.7%) in the 20 min following injection (Fig. 2a shows group-average time-series).

We additionally compared network-level[38] control energies (see "Network-level control energy analyses" section for details), finding reductions under DMT were most prominent in the visual, frontoparietal, and default mode networks (SI Fig. 1). DMT's effects on lowering control energy were strongest in the first half of the post-injection period in the frontoparietal and default mode networks compared to the second half, while the opposite was true for the visual network (SI Fig. 2).

### Global control energy under DMT negatively correlates with signal diversity from simultaneous EEG and subjective drug intensity ratings

We next relate the dynamical changes observed under DMT to signal diversity, quantified in terms of Lempel-Ziv complexity ($LZ_c$), derived from simultaneous EEG recordings (see SI Fig. 3 for a between-condition comparison of $LZ_c$). We correlate the between-condition differences in these metrics (see "Global control energy analyses" section) and find that the more fMRI-based global control energy is decreased under DMT, the more signal diversity from EEG is increased (Fig. 2b; Spearman's $R = -0.35$, $p_{perm} < 0.0001$). Correlating the between-condition differences in control energy and subjective drug intensity (see "Global control energy analyses" section for details), we find that reduction of control energy by DMT correlates with the intensity of the drug's subjective effects over time (Fig. 2c; Spearman's $R = -0.42$, $p_{perm} = 0.0166$). Recalculating each correlation using only the 20-min post-administration period reveals a significant negative correlation between control energy and signal diversity (Spearman's $R = -0.11$, $p_{perm} = 0.0038$) and a non-significant correlation between control energy and intensity (Spearman's $R = 0.28$, $p_{perm} = 0.8926$). The lack of correlation in the latter case is likely due to the fact that because intensity ratings were collected at the end of each minute (i.e., at the end of the last minute prior to DMT injection, and then again one minute after injection), (1) there are fewer sampled points on which to capture the relationship, so it will be noisier, and (2) the window of the largest change in intensity and control energy (right after administration) is not well sampled. We additionally recalculate the main correlations from Fig. 2 using group-averaged framewise displacement values as a covariate of non-interest (Spearman's $R = -0.38$, $p_{perm} < 0.0001$ and Spearman's $R = -0.52$, $p_{perm} = 0.0041$, respectively). Group-level correlations were also performed for each condition separately (SI Fig. 4), showing significant inverse relationships between control energy and signal diversity and intensity in both the DMT and placebo conditions; however, the relationships were stronger in the DMT condition. We additionally performed correlations between control energy and signal diversity and intensity at the subject level (SI Fig. 5–8), finding results consistent with the group-level correlations and generally stronger correlations in the DMT condition than the placebo condition.

### Spatial patterns of control energy and its correlation with signal diversity and intensity are associated with spatial patterns of serotonin 2a receptors

Next, we interrogate control energy specifically under DMT, taking advantage of our in-scanner drug-delivery design to accurately measure (and later simulate) DMT's effects in real time. We start by investigating control energy under DMT at the regional, rather than global, level in order to understand spatial relationships. Regional control energy reflects the amount of input injected into each region in order to complete a desired transition, whereas global control energy is the sum of this metric over all regions. While global control energy provides a single metric useful for summarizing the relative difficulty of state transitions, regional control energy can help quantify the varied contributions of each brain region to the

global measure. We evaluated regional control energy under DMT in three ways (Fig. 3a): (1) the change in regional control energy post-injection relative to pre-injection, (2) each region's control energy correlated with global EEG signal diversity over time during DMT scans, and (3) each region's control energy correlated with subjective drug intensity over time during DMT scans (see "Regional control energy analyses" section for details). Reflecting the global result in Fig. 2, we find that regional control energy is generally decreased following DMT injection, and is generally inversely correlated with both EEG signal diversity and drug intensity ratings (Fig. 3a). Our purpose in assessing control energy at a regional level is to relate its spatial distribution to biologically relevant patterns in the human brain. To do so, we calculated the Spearman rank correlation between each of these regional metrics and the serotonin 2a receptor cortical density distribution derived from PET imaging (Fig. 3b)[39]. To assess the robustness of these regional metrics' correlations with the 2a cortical map, we compared their values against 10,000 correlations calculated with spun permuted[40] 2a cortical maps that preserve spatial autocorrelations present in the original

maps. We find that each of our regional metrics from Fig. 3a is negatively correlated with the serotonin 2a spatial distribution. These negative correlations indicate that brain regions with more serotonin 2a receptors have their control energies decreased more by DMT, and their control energies over time are more negatively correlated with EEG signal diversity and subjective drug intensity (Fig. 3c). See SI Fig. 9 for a replication of the above analyses using the placebo scans.

Through dominance analysis, we next compare the strength of the regional metrics' associations with serotonin 2a receptor maps against their associations with other serotonin receptor subtypes, namely the serotonin (5-HT) 1a, 5-HT1b, and 5-HT4 receptors, and the serotonin transporter (5-HTT). Dominance analysis is a method of ranking the importance of multiple input variables (in our case, the five different serotonin receptor/transporters) in explaining a target variable through a series of linear regression models[41]. A separate dominance analysis was run with each of the three metrics from Fig. 3a as the target variable (see "Dominance analysis" section for details). We find that serotonin 2a density is the most dominant

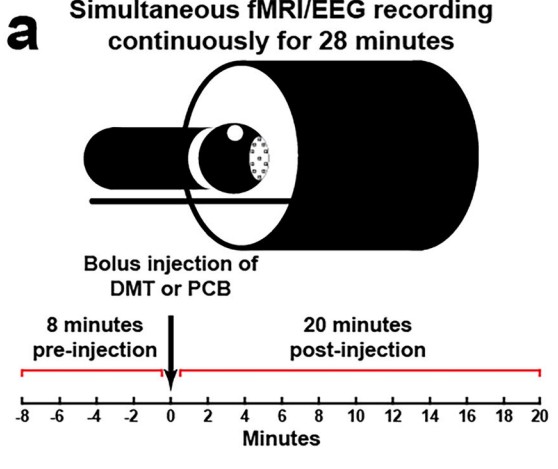

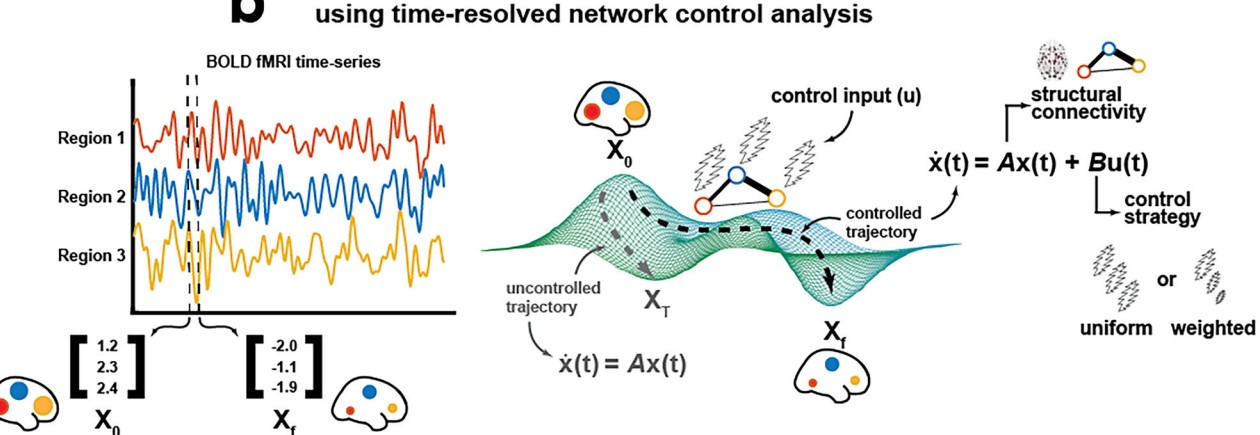

**Fig. 1 | Time-resolved network control analysis of the human brain during a pharmacologically induced alteration of consciousness. a** Fourteen individuals were scanned twice per day on two different days (two weeks apart), receiving either DMT or saline placebo at each of these separate days in a single-blind, counter-balanced design (see "Participants and study design" section for details). On each day, a 28-min-long eyes-closed resting-state EEG-fMRI scan was performed with DMT/placebo intravenously administered at the end of the 8th minute. On the same day, identical scanning sessions were performed where participants were asked to rate the subjective intensity of drug effects at the end of every minute. **b** Here, we deploy a time-resolved network control analysis of the brain's trajectory through its activational landscape. The position in the landscape is illustrated here as a 3D vector containing regional BOLD signal amplitude at a given time $t$. We compute a control

energy time-series from the regional activity vector time-series by modeling transitions between adjacent regional activity vectors ($x_0$ and $x_f$, respectively) using a linear time-invariant model within a network control theory framework. In this framework, the state of the network $x(t)$, here a vector of regional BOLD activations at time $t$, evolves over time via diffusion through the brain's weighted structural connectome $A$, the adjacency matrix. In order to complete the desired transition from the initial ($x_0$) to the target state ($x_f$), input ($u$) is injected into each region in the network. Varying control strategies (reflected in the matrix $B$) may be deployed, wherein different regions are assigned varied amounts of control within the system. Integrating input $u(t)$ at each node over the length of the trajectory from $x_0$ to $x_f$ yields region-wise control energy, and summing over all regions yields a global value of control energy required to complete the transition.

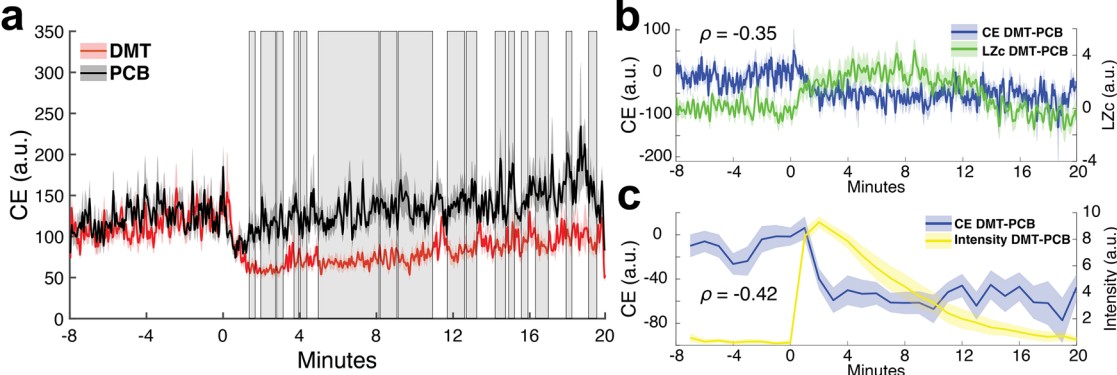

**Fig. 2 | Global control energy is reduced after DMT injection compared to after placebo injection, and negatively correlates with signal diversity and subjective drug intensity ratings. a** Group-average global control energy time-series for the DMT and placebo (PCB) conditions. Nearly two-thirds of post-injection control energies (61.7%) were found to be significantly lower under DMT compared to placebo ($n = 14$ subjects; gray boxes reflect cluster-corrected significant time-points; see "Global control energy analyses" section for details). **b** Differences in global control energy and EEG signal diversity between the DMT and PCB conditions are negatively correlated over the 28 min scans ($n = 838$ time-points; Spearman's $R = -0.35$, $p_{perm} < 0.0001$), indicating that lower demand for fMRI-based global control energy was associated with increased EEG-based signal diversity of brain activity. **c** Differences in global control energy between the DMT and PCB conditions were averaged over one-minute intervals in order to compare with subjective drug intensity ratings (0–10) collected at the end of each minute (the latter of which were obtained during a separate fMRI from the one used to calculate global control energy). We found a negative correlation over time between intensity ratings and differences between the DMT and placebo conditions' global control energies ($n = 28$ time-points; Spearman's $R = -0.42$, $p_{perm}$ 0.0166). Solid lines are group means and corresponding shaded boundaries reflect the standard error of the mean (SEM).

variable when explaining the variance in all three regional metrics (Fig. 4). Notably when replicating this approach using the placebo scans, the fingerprint of receptor dominance varies drastically for each metric (SI Fig. 10).

### DMT's impacts on control energy can be simulated from pharmacokinetics and the serotonin 2a receptor maps

For our simulation of DMT's impact on control energy, we begin with all participant's placebo fMRI scans. Prior to DMT injection, our control strategy, $B$, is uniform. Thus, pre-injection, our simulation matches the placebo control energy from Fig. 2a. Following injection, we begin adding control to the system in a time and space-dependent manner according to our pharmacologically-derived time-varying control strategy (Fig. 5, top; see "Simulating DMT's impacts on control energy" section for details). This strategy successfully estimates DMT's impact on global control energy during the 20 min post-injection (Fig. 5, bottom).

### Supplemental analyses

We performed several analyses to supplement our main findings. First, we reproduce our main results without the use of global signal regression during fMRI preprocessing (SI Fig. 11). We find that these results are consistent with the main text; however, the regional metrics are less varied and have weaker correlations with 2a receptor maps. We also show scatter plots as in Fig. 3, but with subcortical regions included (SI Fig. 12). The corresponding correlations are all strong and negative; however, spin testing was not performed to calculate the p-values, as subcortical regions cannot be spin permuted. In SI Fig. 13, we show distance metrics between true and simulated DMT control energies for two alternative models to demonstrate the advantage of incorporating both the simulated brain-effect concentration (compared to simulated plasma concentration) and the spatial 2a receptor density map (compared to a uniform spatial map). Lastly, we examine DMT's impacts on global control energy under a variety of BOLD signal normalization approaches (SI Fig. 14). All methods of activity normalization result in a significant decrease in control energy after DMT injection, except for L2 normalization, which results in a significant increase in control energy. This suggests that when accounting for the magnitude of activity, states are more difficult to reach through a network control process. This is likely related to the concurrent psychedelic effect of decreased signal magnitude and increased temporal entropy. When magnitude is normalized and its effects removed from the energy calculations, the decreased autocorrelation between adjacent BOLD activity dominates the energy

calculations and results in increased control energy. When both magnitude and inter-state distance are accounted for (double normalization), DMT still decreases control energy. This suggests that when both factors are accounted for, states are still closer together through a network diffusion process after DMT compared to before DMT.

### Discussion

In this work, we use a time-resolved network control theory framework to characterize brain activation dynamics underlying the DMT psychedelic experience (Fig. 1). We find that, on a global level, control energy is reduced under DMT compared to placebo. We then interrogate the effect of DMT in a temporally- and spatially-resolved manner. Temporally, we find that the decrease in global control energy under DMT covaries with increases in EEG signal diversity and subjective drug intensity (Fig. 2). Spatially, we find that the regional distribution of DMT's effects is aligned with serotonin 2a receptor densities (Figs. 3 and 4). Finally, we demonstrate a computational framework for predicting DMT's impact on global brain dynamics using a network control model informed by pharmacokinetics and pharmacodynamics (Fig. 5).

Comparing global control energies between DMT and placebo reveals an overall reduction in the input necessary for transitioning between brain-states following injection with DMT (Fig. 2a). Prior work with other psychedelics (LSD and psilocybin) has demonstrated that these serotonergic 2a agonist compounds increase the diversity of brain-state dynamics[15,42–47]. Decreased control energy may be reflective of a system poised near a state of criticality, whereby lowered barriers facilitate access to a larger repertoire of state dynamics[10,16,48]. In our own recent work using an approach similar to the one employed here, we found that LSD and psilocybin decreased control energy, and, across individuals, larger decreases under LSD were associated with more complex brain-state sequences extracted from fMRI[24]. Other recent studies have also found associations across individuals between control energy and fMRI-based entropy or diversity metrics[33,49]. The present study design enables us to further strengthen the association between control energy and neural entropy by showing that the global control energy throughout the fMRI time-series is temporally coupled with neural signal diversity measured with simultaneous EEG, the latter of which does not depend on neurovascular coupling which are likely altered under psychedelics[50] (Fig. 2b). Signal diversity here refers to the Lempel-Ziv complexity of EEG signal averaged across electrodes, and its increase has been a consistent characteristic of acute psychedelic experiences[7,47]. We also

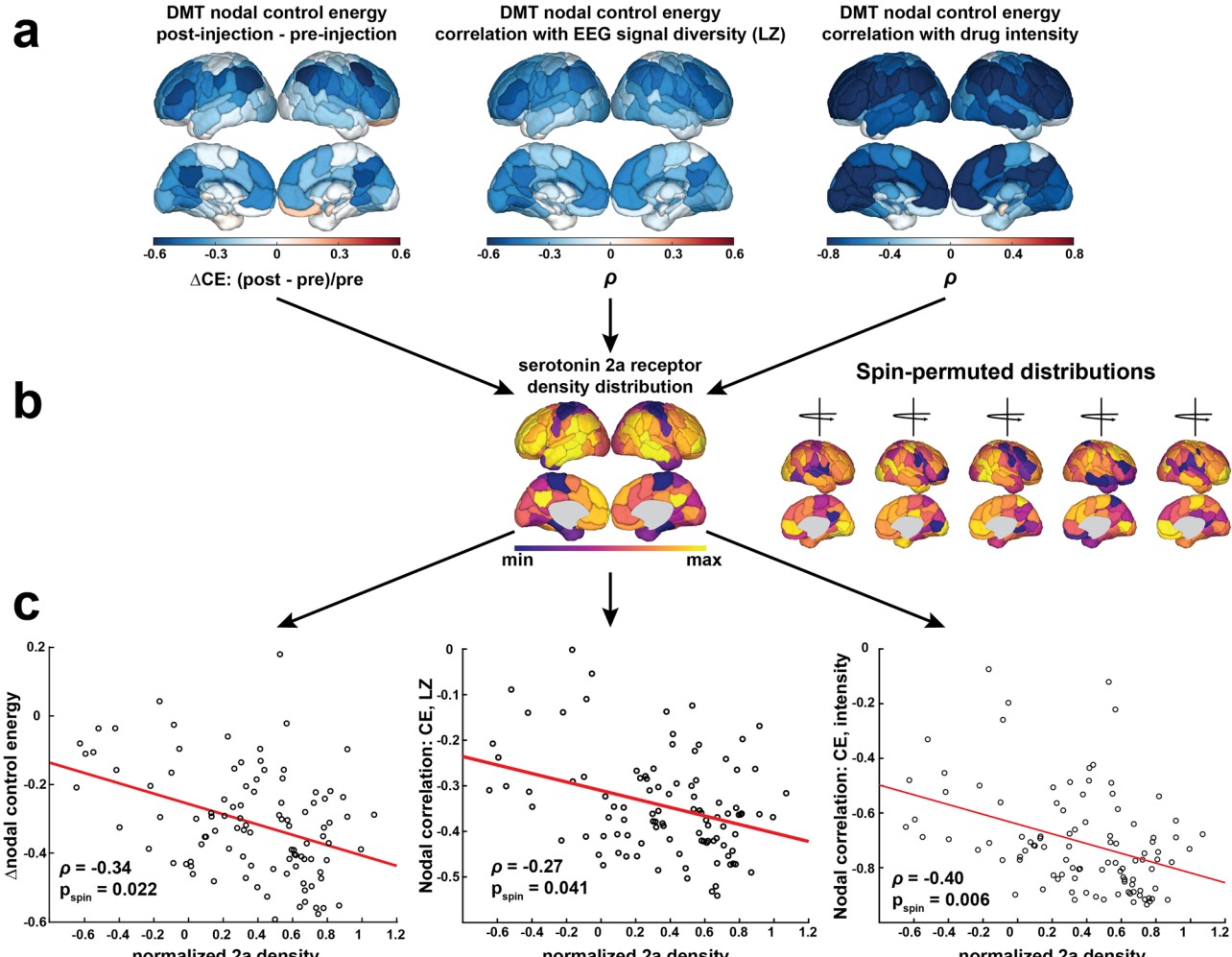

**Fig. 3 | Regional control energy and its temporal correlation with signal diversity and drug intensity are associated with serotonin 2a receptor maps. a** Regional control energy metrics. (left) The change in regional control energy in the 8 min after DMT injection, relative to the 8 min prior to the injection. (middle) Each region's control energy time-series over the course of the full 28 min DMT scans correlated with global signal diversity from EEG during the same scans. (right) Regional control energy during the DMT scans was averaged over one-minute windows corresponding to the timing of subjective drug intensity ratings from separate scans. The windowed control energy time-series for each region was then correlated with the subjective drug intensity ratings. **b** Each of the regional metrics in (**a**) were then correlated with the cortical spatial map of the serotonin 2a receptor derived from PET[39]. The strength of these correlations were compared against null correlations with 10,000 cortical spin permutations[40] of the 2a receptor map. **c** Scatter plots of the three cortical regions' metrics (*n* = 100 regions) from (**a**) and serotonin 2a receptor density from (**b**).

find that control energy over time is inversely related to the intensity of the subjective drug effects (Fig. 2c), linking our fMRI-based metric to participants' experiences in real time.

On a network level, DMT exerted its effects most strongly on the visual, frontoparietal, and default mode networks (SI Fig. 1). DMT, and other psychedelics more broadly, are known to preferentially impact all three of these networks[51], albeit findings in the visual network have been less consistent. Our time-resolved design may shed some light on this inconsistency. While DMT's effects on the control energy of the frontoparietal and default mode networks are strongest during the first half of the post-injection period, the opposite is true for the visual network (SI Fig. 2). Notably, DMT's large effects in the visual network in the second half of the post-injection scanning period appear to be due to control energy gradually increasing throughout the post-injection scanning period in the placebo condition, while low-levels are maintained in the DMT condition throughout. This may in part be an arousal effect. BOLD amplitude is known to fluctuate with the degree of wakefulness[52], and BOLD amplitude in the visual cortex in particular increases as arousal drops and individuals transition to light sleep[53]. We speculate that arousal under the placebo and DMT conditions is high around the injection period, and remains high in the DMT condition

but slowly decreases over time in the placebo condition, leading to increases in control energy for the visual network under placebo and a maintained reduction in control energy under DMT.

We next sought to interrogate regional differences in DMT's effects on control energy and its association with EEG signal diversity and subjective intensity. In general, the regional metrics reflect what is observed at a global level - namely, regional control energy is decreased following injection with DMT and is inversely correlated with EEG signal diversity and subjective drug intensity (Fig. 3a). Of particular interest to us was the extent to which regional heterogeneity in these metrics might be explained by biologically relevant information. The serotonin 2a receptor is the primary target in the brain responsible for initiating a cascade of changes that give rise to the characteristic subjective and neural effects of psychedelics[18–20]. Our previous simulation studies demonstrated that the serotonin 2a receptor spatial map was particularly suited for lowering global control energies[24]. Here, we further demonstrate that regional differences in empirical control energies during psychedelic administration are related to regional serotonin 2a receptor densities (Fig. 3c). We rank-correlated each of the three regional metrics' values with that of the serotonin 2a cortical distribution. In each case, we find that the regional spatial pattern of control energy differences

**Fig. 4 | Dominance analysis reveals the highest relative importance of the serotonin 2a receptor in DMT-related changes in cortical activity metrics.** Three separate dominance analyses were performed using cortical values from five PET-derived serotonin receptor and transporter spatial densities[39] as input variables and each cortical metric from Fig. 3a as the output. Dominance analysis assesses the relative importance of each input in explaining the output variable's variance while controlling for the contributions of other predictors in multiple regressions[41]. Displayed is the percent relative importance given to each receptor/transporter map for explaining the variance in each cortical metric, as determined by dominance analysis. $n = 116$ regions; 5-HT = serotonin (5-hydroxytryptamine).

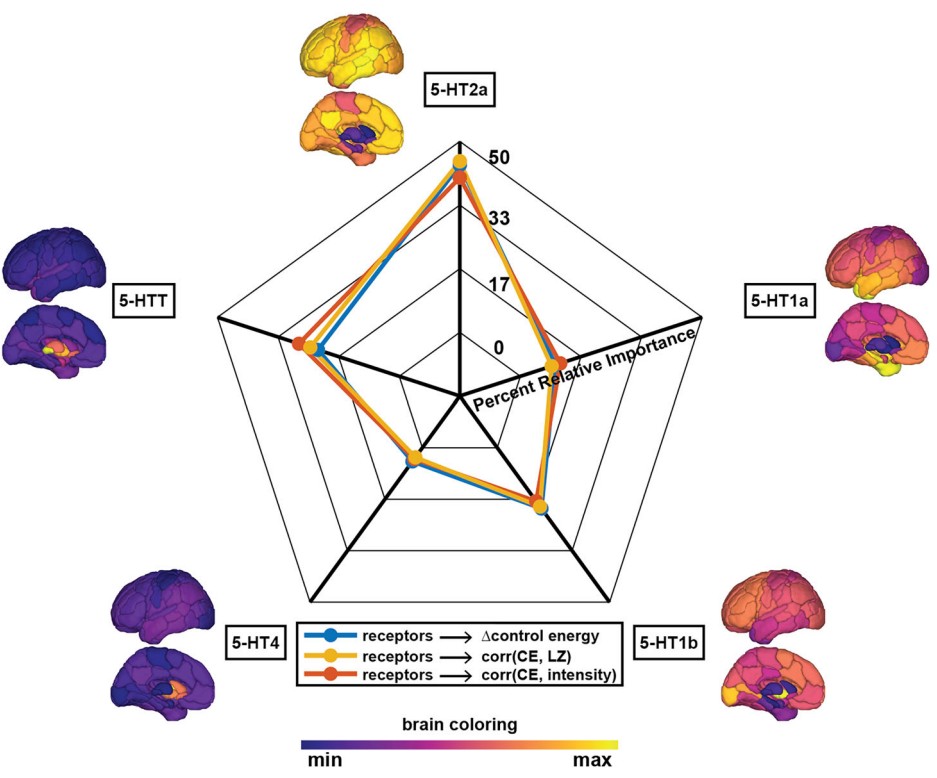

and its association with signal diversity and drug intensity are inversely related to the density of the serotonin 2a receptor (Fig. 3c). In recent years, there has been increasing appreciation that brain maps exhibit spatial autocorrelation: in other words, whereas correlation assumes that data-points are independent, brain maps violate this assumption because two neighboring regions will systematically exhibit values that are more similar, than two regions that are far apart. In other words, similarity of regional values is influenced by spatial proximity. In turn, this can result in an inflated false positive rate: two random brain maps can still appear significantly spatially correlated, just as a result of spatial autocorrelation. One popular method to account for this is to use so-called spin-nulls[54]: projecting one of the maps on a sphere, then rotating the sphere randomly, and re-projecting the spun data onto the brain. The resulting spun map will have the same spatial autocorrelation as the original map, but allocated randomly in terms of neuroanatomy: therefore, its correlation with the map of interest will be purely due to spatial autocorrelation. By building a null distribution of such maps, one embodies the null hypothesis that the correlation between maps of interest is only due to spatial autocorrelation. Here, we compared the strength of these correlations against a distribution of spin-nulls[40] to demonstrate significance above and beyond the effect of spatial autocorrelation.

One might ask, however, whether the serotonin 2a receptor is associated with the regional metrics at a level above and beyond other serotonin system receptors. To answer this question, we performed a dominance analysis[41] for each regional metric using five input variables: the serotonin 2a (5-HT2a), serotonin 1a (5-HT1a), serotonin 1b (5-HT1b), serotonin 4 (5-HT4) receptors, and the serotonin transporter (5-HTT) (see "Dominance analysis" section for details). The serotonin 2a receptor was found to have the highest relative importance in explaining the variance of regional decreases in control energy after DMT, as well as the control energy's correlation with signal diversity and subjective intensity (Fig. 4). In general, the dominance maps for each of the three metrics yielded a consistent fingerprint across receptors. In contrast, when we replicated this analysis using the placebo condition (SI Fig. 10), there was no consistent finger-printing across metrics, and the 2a receptor was dominant only for control energy reduction. Together, these results suggest that regions having higher

densities of the serotonin 2a receptor are the most impacted by DMT, and have stronger couplings with neural and subjective effects. A recent dual-receptor model of psychedelic action positions the serotonin 1a receptor as an influential modulator of their effects[55]. Compared with other classic psychedelics like LSD and psilocybin, DMT has a higher serotonin 2a:1a receptor binding ratio[56]. While, as discussed above, the serotonin 2a receptor is the primary receptor responsible for much of psychedelics' neural and subjective effects, their action at other serotonin receptors is of increasing interest to the field. Indeed, rodent models have suggested that some of psychedelics' antidepressant and synaptogenic effects may act through non-2a serotonergic mechanisms[57,58]. Among other serotonin receptors considered here after the 2a receptor, DMT's effects were best explained by the serotonin 1b receptor, closely followed by the 1a receptor. This finding is consistent with our previous work on LSD and psilocybin[24].

Having established an association between DMT's impact on control energy and the serotonin 2a receptor distribution, we finally ask the question: "can DMT's impact on control energy be simulated from non-drug data (i.e., in this case, the placebo dataset) using a pharmacologically-informed control framework"? Our control energy calculations up to this point were agnostic to regional heterogeneity, i.e., they deployed a uniform control strategy (encoded by the control matrix $B$ being the identity). However, adjustments to the control strategy have been successfully deployed for in silico hypothesis testing[26,27,30,31] and simulations of external and internal forms of stimulation[24,29,35]. DMT injection exhibits rapid onset of subjective and neural effects in a concentration-dependent manner[8,9]. Previous work has demonstrated successful pharmacokinetic modeling of DMT plasma concentration and its neural effects at dosing regimens similar to those used in the present study[59,60]. Here, we used an independently validated model of DMT's pharmacokinetic impact on EEG alpha rhythms to simulate population-level 'brain-effect' concentrations of DMT over the course of the 28 min scans[60]. We next combine this temporal estimation of DMT's effects with spatial information by multiplying the simulated DMT concentration at each time-point with the regional serotonin 2a receptor maps[39]. This yields a temporal and spatial map of DMT's hypothesized impact on brain dynamics, which we operationalize as a time-varying control strategy within our network control theory approach (Fig. 5). Our

**Fig. 5 | Global control energy time-series for the DMT condition is simulated using only placebo fMRI data, coupled with simulated DMT plasma concentrations and 2a receptor density information.** Pharmacokinetic modeling yields an estimate of DMT concentration over the length of the 28 min scans. Here, we specifically used predicted 'brain-effect compartment' concentrations from a previously validated model using plasma concentration sampling and EEG[60]. Multiplying DMT concentration over time by regional PET-derived serotonin 2a densities[39] yields an estimate for DMT's impact on each brain region over time, which can be used as a time-varying control strategy. In order to simulate the impact of DMT on the global control energy time-series, we use each participant's placebo fMRI data and apply the time-varying control strategy via inclusion in the diagonal of matrix *B*. Prior to DMT injection, the control strategy (diagonal in *B*) is uniform, as is the case for all previously calculated energy metrics. Solid lines are group means, and corresponding shaded boundaries reflect the standard error of the mean (SEM). CE = control energy; a.u. = arbitrary units.

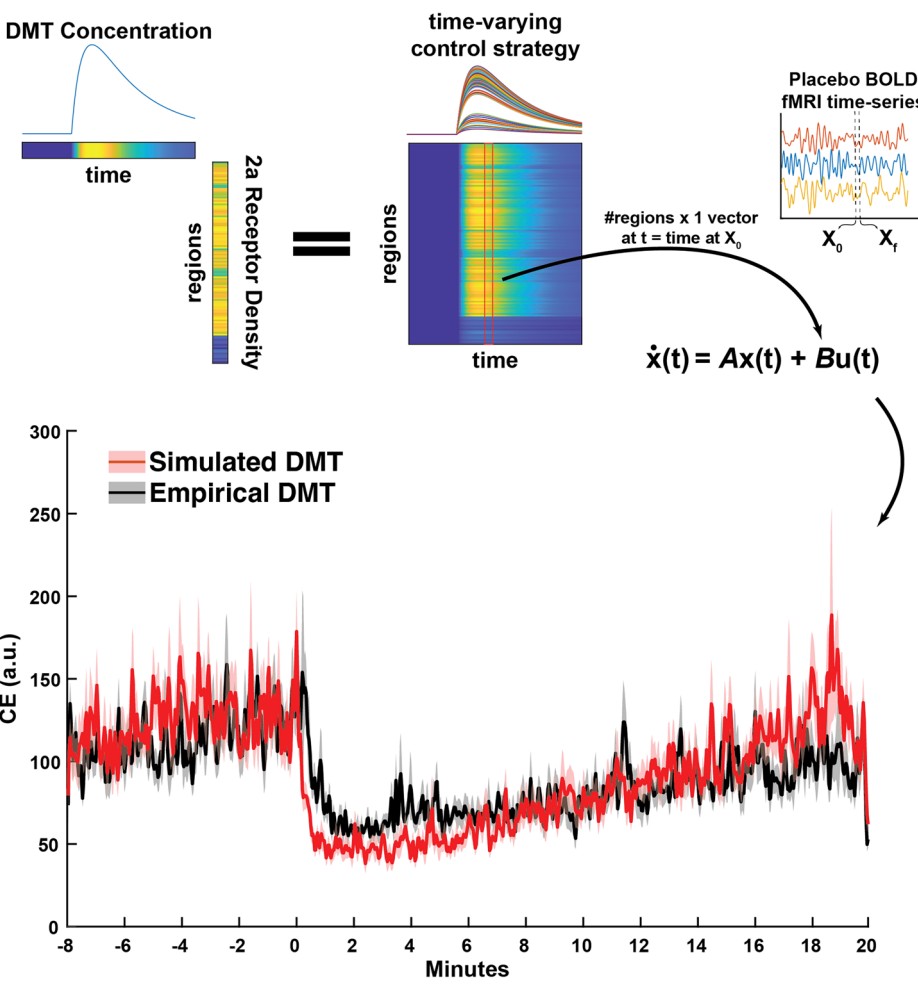

ability to estimate DMT's impacts on control energy through pharmacologically-informed adjustments to model parameters serves as an important proof-of-concept for using network control theory in computational psychiatry applications[61,62]. Importantly, we validated the superior utility of our two specific pieces of information in the control strategy - the 'brain-effect' versus the plasma concentration and the 2a receptor's spatial information versus a uniform control approach (SI Fig. 13).

The present work validates a meaningful fMRI correlate of the increased signal complexity or entropy that has been reliably demonstrated with EEG or MEG recordings of the psychedelic state[7,47,60,63,64]. The latter modalities have limited spatial resolution (EEG) or depth (MEG), precluding inferences about specific regional effects or changes in deep structures. Thus, observing fMRI correlates of other modalities' recordings deepens our understanding of the so-called 'entropic brain' effect[36,43]. The strength of fMRI is its high spatial resolution and whole-brain coverage; however, relative to EEG, its cost, immobility, and other practical challenges limit its widespread application. Recent work has shown that EEG-recorded signal complexity or entropy can be effectively tracked in real time[65], inspiring ideas regarding how such information could be used to adapt treatment parameters such as dosage, in a way to suit individual responses. More work across all modalities is required to further deepen our understanding of mechanisms of psychedelic action, e.g., to test whether the observed acute brain effects begin with a 5-HT2a receptor agonism initiated spike-to-wave decoupling[66] translating downstream into anatomical and functional neuroplastic changes such as those found in preclinical[21,57,58,67,68] and clinical neuroscience research[69,70].

Although the present analysis expands on prior work in this field, several important limitations must be considered. First, our small sample size limits the generalizability of our findings; however, this is the third independent dataset (including one each in LSD and psilocybin[24]) that has found decreased control energy during psychedelic administration, which lends more confidence in our findings. Our time-resolved control energy analysis provides the ability to associate control energy changes in real time with other imaging metrics and subjective effects. However, like many time-resolved metrics, this increases noise sensitivity. For our correlational analyses, we consider associations between group-level metrics in order to more accurately assess changes over time and space. In addition, we assess our simulation's success based on the group-average outputs from our model. Extending the work toward individual-level modeling will be a fruitful avenue for future research, with implications for personalized medicine[61,62]. Elucidating the potential impacts of vasoconstriction[50,71] and arousal[52] on our findings also requires sophisticated experimental designs (e.g., using an active stimulant as control) and should be investigated in future studies. Additionally, here, as in our prior work with LSD and psilocybin, we relied on group-averaged structural connectomes from the HCP for controllability analysis. The utilization of individualized networks would allow for additional analyses. While participant-level diffusion data is not always readily available, one fruitful avenue for future research in psychedelics may be to perform controllability analyses on participant-level whole-brain networks generated through functional or effective connectivity – whereby one could study how the drastic functional reorganization under psychedelics may facilitate or hinder various information pathways in the brain. Lastly, a single-blind design was implemented. Although blinding has pragmatic challenges in psychedelic research[72,73], we observed in previous work that cardiac activity increased at the time of injection in both the placebo and DMT conditions, aiding in the control of arousal confounders. However, future work could benefit from the assessment of expectancy regarding the psychedelic experience.

In summary, we have demonstrated that time-resolved network control analysis captures meaningful changes in brain activation dynamics during the onset, peak, and offset of an infused DMT experience. We found significant reductions in the control energy required for the brain to traverse its activity landscape under DMT, and an association between decreases in control energy and increases in EEG-based neural signal complexity and subjective drug effects in a manner that is regionally related to serotonin 2a receptor density. Finally, we demonstrated that through a pharmacologically-informed network control framework, we are able to simulate DMT's impacts on control energy over time - an important step towards understanding mechanisms of these neuromodulators.

## Methods

### Participants and study design

The original single blind (only researchers were aware of the order of administration), placebo controlled, counterbalanced study was approved by the National Research Ethics (NRES) Committee London – Brent and the Health Research Authority and was conducted under the guidelines of the revised Declaration of Helsinki (2000), the International Committee on Harmonisation Good Clinical Practices guidelines, and the National Health Service Research Governance Framework. Imperial College London sponsored the research, which was conducted under a Home Office license for research with Schedule 1 drugs. Exclusion criteria included: <18 years old at the moment of participation, MR contraindications, absence of experience with a psychedelic, an adverse reaction to a psychedelic, history of psychiatric or physical illness rendering unsuitable for participation (i.e., diabetes, epilepsy, or heart disease), family history of psychotic disorder, or excessive use of alcohol or drugs of abuse. All ethical regulations relevant to human research participants were followed.

Volunteers participated in two testing days, separated by 2 weeks. On each testing day, participants arrived and were tested for drug abuse and were involved in 2 separate scanning sessions. In this initial session (task-free) they received intravenous (IV) administration of either placebo (10 mL of saline) or 20 mg DMT (in fumarate form dissolved in 10 mL of saline)-injected over 30 s, and then flushed with 10 mL of saline over 15s-in a counterbalanced order (half of the participants received placebo in the first scanning session and DMT in the second scanning session, and the other half received DMT in the first scanning session and placebo in the second scanning session). The first session always consisted of continuous resting-state scans, which lasted 28 min with DMT/placebo administered at the end of the 8th minute. Participants lay in the scanner with their eyes closed (an eye mask was used to prevent their eyes from opening), while EEG activity was recorded. The second scanning session then followed the first after participants had fully returned to baseline (at least two hours later) with the same procedure as the initial session, except on this occasion, participants were asked to rate the subjective intensity of drug effects every minute in real time. Specifically, the ratings were collected by playing an audio cue ("Intensity from 0 to 10") through MR-compatible earphones, which was followed by the participants' verbal response collected via an MR-compatible microphone. On the second testing day, participants who received a placebo for the first scanning session on the first testing day then received DMT for the first scanning session, and vice versa. Due to the differential tolerance of biological and subjective effects to DMT in humans[74], carryover effects from one testing day to the other are not expected, and the subjective intensity experienced for the same dose is expected to be roughly similar between testing days.

This article only analyzes the resting-state scans in which no intensity ratings were asked, but uses the intensity ratings for correlational analyses. In total, 20 participants completed all study visits (7 female, mean age = 33.5 years, SD = 7.9).

### EEG-fMRI acquisition

Images were acquired in a 3T MRI (Siemens Magnetom Verio syngo MR B17) using a 12-channel head coil for compatibility with EEG acquisition. Functional imaging was performed using a T2*-weighted BOLD sensitive gradient echo planar imaging sequence with the following parameters: repetition time (TR) = 2000 ms, echo time (TE) = 30 ms, acquisition time (TA) = 28.06 min, flip angle (FA) = 80°, voxel size = $3.0 \times 3.0 \times 3.0$ mm$^3$, 35 slices, interslice distance = 0 mm. Whole-brain T1-weighted structural images were also acquired.

EEG was recorded inside the MRI during image acquisition at 31 scalp sites following the 10–20 convention with an MR-compatible BrainAmp MR amplifier (Brain Products, Munich, Germany) and an MR-compatible cap (BrainCap MR; Brain Products GmbH, Munich, Germany). This system referenced all electrodes to FCz, and AFz served as the ground electrode. Two additional ECG channels were used to improve heart rate acquisition for artifact minimization during EEG preprocessing, and all impedances were kept below 20 kΩ. EEG was sampled at 5 kHz and with a hardware 250 Hz low-pass filter. EEG-MR clock synchronization was ensured using the Brain Products SyncBox hardware. Additional recordings of 5 min eyes-closed resting-state were performed outside the scanner before DMT/placebo was administered in order to determine the profile of EEG activity in the time and frequency domain and ensure that artifact minimization procedures achieved a similar profile.

### fMRI preprocessing

The same preprocessing pipeline as used in previous work with LSD and psilocybin[24] and reported in Timmermann and colleagues[37] was used here. Scrubbing and nuisance regression was used, which has been shown to be an effective denoising strategy in part due to the removal of high-motion individuals[75]. Six out of 20 participants were discarded from group analyses due to excessive head movement during the 28 min DMT scans (>20% of scrubbed volumes with a scrubbing threshold of framewise displacement (FD) of 0.4[76]), leaving 14 for analysis. Preprocessing steps consisted of (1) despiking (3dDespike, AFNI[77]), (2) slice time correction (3dTshift, AFNI), (3) motion correction (3dvolreg, AFNI) by registering each volume to the most similar volume, in the least squares sense, to all others (minimizing the amount of transformation required for each volume), (4) brain extraction (BET, FSL[78]), (5) rigid body registration to anatomical scans, (6) non-linear registration to 2 mm MNI brain (Symmetric Normalization (SyN), ANTS[79]), (7) scrubbing - using an FD threshold of 0.4 - with scrubbed volumes being replaced with the mean of the surrounding volumes. Additional preprocessing steps included: (8) spatial smoothing (FWHM) of 6 mm (3dBlurInMask, AFNI), (9) band-pass filtering between 0.01 and 0.08 Hz (3dFourier, AFNI), (10) linear and quadratic de-trending (3dDetrend, AFNI), (11) regressing out 9 nuisance regressors (all nuisance regressors were band-pass filtered with the same filter as in step 9), out of these, 6 were motion-related (3 translations, 3 rotations) and 3 were anatomically-related. Specifically, the anatomical nuisance regressors were: (a) ventricles (Freesurfer[80], eroded in 2 mm space), (b) draining veins (FSL's CSF minus Freesurfer's ventricles, eroded in 1 mm space[78,80]), and (c) local white matter (WM) (FSL's WM minus Freesurfer's subcortical gray matter structures, eroded in 2 mm space[78,80]). Lastly, global signal regression was performed, and time-series were parcellated into 100 cortical[81] and 16 subcortical[82] regions of interest. Because global signal regression remains a hotly debated topic, particularly in the context of altered states of consciousness[83–85], we repeated our main analyses without the use of global signal regression (SI Fig. 11).

### EEG preprocessing and signal diversity calculation

Gradient artifacts (GA) caused by the fMRI were removed using an average artifact template subtraction (AAS) algorithm, which is part of the Brain-Vision Analyser software. The algorithm computes a representative template artifact based on a sliding average of 21 TR windows, which is then subtracted from each TR window, thereby removing most of the MR-related noise[86]. Following gradient artifact correction, the data were downsampled to 250 Hz, and ballistocardiogram (BCG) artifacts were reduced by placing heartbeat markers corresponding to the R-peak determined on a representative template of the signal corresponding to the ECG channels (low-pass filtered at 15 Hz). An AAS algorithm was used to correct for the pulse

artifact by producing a template resulting from averaging multiple cardiac cycles using a sliding-window approach, generating a different template for each sliding window, which is subtracted from that period[87]. The following preprocessing steps were performed using the Fieldtrip software[88]: The data were demeaned, band-pass filtered at 1–45 Hz, and epoched in separate 2-s trials. The data were then visually inspected, and trials containing artifacts associated with jaw clenches and gross artifacts were removed. Independent component analyses were subsequently performed to remove residual BCG and GA artifacts as well as eye movements. If remaining gross artifacts were observed, the corresponding segments of the data were removed, and ICA was run again for improved results. Validation of preprocessing results was performed by comparing out-of-scanner EEG profiles in the time and frequency domains, as well as in-scanner data.

Following our previous study involving DMT[7], as well as those performed with LSD, psilocybin and ketamine[47], we performed signal diversity analysis using the Lempel-Ziv 1976 algorithm (LZ76), as reported in Timmermann et al.[37]. The EEG signal at each single electrode was binarized using its mean for each 2-s epoch, and then the LZ76 algorithm was used to generate a dictionary of unique subsequences whose size quantifies the temporal diversity for the signal (denoted here as LZs). The average LZs across channels were convolved with a hemodynamic response function and then used for correlational analyses with control energy.

## Structural connectivity network construction

The structural connectome used for network control theory analysis was identical to the one used previously to be consistent with our prior work in this area[24,45]. Namely, we relied on diffusion data from the Human Connectome Project (HCP, http://www.humanconnectome.org/), specifically from 1021 subjects in the 1200-subject release[89]. A population-average structural connectome was constructed and made publicly available by Yeh and colleagues[90] in the following way. Multishell diffusion MRI was acquired using b-values of 1000, 2000, 3000 s/mm$^2$, each with 90 directions and 1.25 mm iso-voxel resolution. Following previous work[24,45,91], we used the QSDR algorithm implemented in DSI Studio (http://dsi-studio.labsolver.org) to coregister the diffusion data to MNI space, using previously adopted parameters[91]. Deterministic tractography with DSI Studio's modified FACT algorithm then generated 1,000,000 streamlines, using the same parameters as in prior work, specifically, angular cutoff of 55°, step size of 1.0 mm, minimum length of 10 mm, maximum length of 400 mm, spin density function smoothing of 0.0, and a QA threshold determined by DWI signal in the CSF. Each of the streamlines generated was screened for its termination location using an automatically generated white matter mask to eliminate streamlines with premature termination in the white matter. Entries in the structural connectome $A_{ij}$ were constructed by counting the number of streamlines connecting every pair of regions $i$ and $j$ in the augmented Schaefer-116 atlas[81,82]. Lastly, the streamline count was normalized by the number of voxels contained in each pair of regions.

## Control energy calculation

Network control theory allows us to probe the constraints of white-matter connectivity on dynamic brain activity, and to calculate the minimum energy required for the brain to transition from one activation pattern to another[24,30,92]. We obtained a representative $N \times N$ structural connectome $A$ obtained as described above using deterministic tractography from HCP subjects (see "Structural connectivity network construction" section), where $N$ is the number of regions in our atlas. We then employ a linear time-invariant model:

$$\dot{x}(t) = Ax(t) + Bu(t) \tag{1}$$

where $x$ is a vector of length N containing the regional activity at time $t$. $B$ is an $N \times N$ matrix that contains the control input weights, and is otherwise known as the control strategy. In our analyses, $B$ is constructed by placing an input vector, $v$, along the diagonal of the identity matrix $B$. In uniform control scenarios (i.e., all analyses except the final

simulation of DMT from placebo data), $v$ is a vector of length N containing all ones.

To compute the minimum control energy required to drive the system (network) from an initial activity pattern ($x_0$) to a final activity pattern ($x_f$) over some finite amount of time ($T$), we minimize the inputs ($u(t)$) subject to Eq. 1:

$$u(t)^* = \min \int_0^T u^\top(t)u(t)\,dt \tag{2}$$

where $T$ is the time horizon that specifies the time over which input to the system is allowed. Here, a common choice of $T = 1$ was used[27,28,92,93]. The minimum control energy for a single brain region $i$ is then the integration of those inputs over the time horizon:

$$E_i^* = \int_0^T \left\| u(t)_i^* \right\|_2^2 dt \tag{3}$$

And, finally, the global minimum control energy for a transition is the sum of Eq. 3 over each region:

$$E_{\min} = \sum_{i=1}^{N} E_i^* \tag{4}$$

For the network-level comparisons in the SI Fig. 1, Eq. 3 was summed over the regions assigned to each Yeo network[38] and the subcortex. Regional, network, and global quantities were calculated for each pair of initial $x_0$ and final $x_f$ brain states (i.e., adjacent BOLD volumes in each individual's fMRI scans) to obtain a time-series of control energy over each individuals' 28 min scanning sessions. The primary analysis was conducted on states obtained directly from the preprocessed fMRI; however control energy was also calculated using a variety of state-normalization techniques[94] comparing the average of global control energy before and 8 min after DMT injection (SI Fig. 14) including: (a) radial normalization, where final states are normalized so that they are unit distance from initial states (b) L2 normalization, where all states are normalized by their L2 magnitudes, (c) distance normalization, where each pair of initial and final states are normalized by the magnitude of their inter-state distance, (d) double normalization, where first L2 normalization is applied to all states, then distance normalization is applied to all pairs of initial and final states prior to control energy calculation.

## Global control energy analyses

To test whether control energy was lower under DMT, we compared the global control energy time-series between DMT and placebo at each transition (Fig. 2a) using cluster-based permutation testing[88] with a significance threshold of α = 0.05. We then performed several correlational analyses using the global control energy time-series and EEG signal complexity. First, to summarize the relationship across both conditions in the main text (Fig. 2b), we correlated the delta (DMT - placebo) of the group-level (mean) control energy time-series with the delta of the group-level EEG signal complexity across the full 28-min time-series (838 transitions). We then repeated this correlation using (a) frame-by-frame displacement as a covariate, and (b) using only the 20-min post-injection scanning period (reported in text). We additionally correlated global control energy with EEG signal complexity separately for each condition (SI Fig. 4), and then again separately for each individual scan (SI Fig. 5–6). In separate scanning sessions with an identical dosing regimen, the same participants rated the subjective intensity of drug effects on a scale of 0–10 at the end of every minute. We performed the same series of correlations, this time averaging control energy over one-minute intervals corresponding to the collection of subjective intensity ratings and correlating this 28-point time-series with the intensity ratings (Fig. 2c; SI Fig. 4, SI Fig. 7–8). All correlations were Spearman rank, and $p$-values were obtained via comparison against 10,000 random permutations. $P$-values were corrected for multiple comparisons[95] for all group-level correlations and subject-level correlations separately.

## Network-level control energy analyses

In-line with the global comparison between DMT and placebo, we also compared the network-level control energy time-series between DMT and placebo at each transition for each of the seven intrinsic functional networks of Yeo and colleagues[38] and an additional subcortical network using cluster-based permutation testing[88] with a significance threshold of α = 0.05 (SI Fig. 1) in order to understand how DMT may preferentially impact some networks over others at different points during the experience. For the most broadly impacted networks (visual, frontoparietal, and default mode), we summarized their changes in the first half and second half of the post-injection period by plotting t-statistics for each transition compared (SI Fig. 2).

## Regional control energy analyses

Having demonstrated DMT's impact on control energy with respect to placebo, and its relationship to EEG signal diversity and subjective intensity ratings, we next sought to understand the regional relationships of control energy specifically under DMT and its relationship to the serotonin 2a receptor's spatial distribution. First, we calculated each region's mean change in control energy during the first 8 min post-injection compared to and adjusted by the region's mean control energy during the 8 min pre-injection window to summarize DMT's peak effects on regional control energy and to ensure the same amount of data was going into both pre- and post-injection metrics (Fig. 3a, left). Next, we correlated each region's mean (over subjects) control energy time-series under DMT (full 28-min scan) with EEG signal complexity (Fig. 3a, middle), and lastly we correlated each region's mean control energy time-series (windowed over 1-min intervals) with subjective intensity ratings (Fig. 3a, right). Each of these metrics was then used as regional spatial maps to compare against the serotonin 2a receptor (see 5-HT receptor mapping for acquisition details). The cortical values of each metric were Spearman rank correlated with the serotonin 2a spatial distribution, and p-values were generated via comparison against 10,000 spin permutations[40] to test for significance above and beyond the effects of spatial autocorrelation in the data (Fig. 3c) and then subsequently corrected for multiple comparisons[95]. These correlations were repeated, including the subcortical values (SI Fig. 12).

## 5-HT receptor mapping

PET data for 210 participants (not under the influence of psychedelics) were acquired on a Siemens HRRT scanner operating in 3D acquisition mode with an approximate in-plane resolution of 2 mm (1.4 mm in the center of the field of view and 2.4 mm in cortex)[96]. Scan time and frame length were designed according to the radiotracer characteristics. All structural MRIs (T1 and T2) were unwarped offline using FreeSurfer's gradient_nonlin_unwarp version 0.8 or online on the scanner. For further details on PET data processing, see Knudsen et al.[97]. The voxelwise average density (Bmax) maps for each receptor were parcellated into 116 regions of interest for the augmented Schaefer-116 atlas[81,82].

## Dominance analysis

Having demonstrated that the regional metrics above were related to the serotonin 2a receptor's spatial distribution, we next sought to determine if this was the case above and beyond their relationships with other serotonin system receptors/transporters. Dominance analysis was used in order to determine the relative importance of each receptor/transporter map in predicting nodal control energy metrics. Dominance analysis aims to establish the relative significance (or "dominance") of each independent variable in relation to the overall fit (adjusted $R^2$) of the multiple linear regression model (https://github.com/dominance-analysis/dominance-analysis)[41]. This process involves fitting the same regression model to every possible combination of predictors (creating $2^p - 1$ submodels for a model with p predictors). Total dominance is characterized as the mean increase in $R^2$ when incorporating a single predictor of interest into a submodel, considering all $2^p - 1$ submodels. The aggregate dominance of all

input variables equals the total adjusted $R^2$ of the comprehensive model, rendering the relative importance percentage an easily understandable technique that apportions the overall effect size among predictors. As a result, in contrast to alternative methods for evaluating predictor significance, such as those based on regression coefficients or univariate correlations, dominance analysis takes into account predictor-predictor interactions and offers interpretability.

## Simulating DMT's impact on control energy

In the case of our DMT simulation, a time-varying control strategy (B) was used (Fig. 5, top), where the input vector, v was a function over time of simulated DMT concentration $DMT(t)$, the regional serotonin 2a receptor density vector ρ, and a scaling parameter α:

$$v(t) = 1 + \alpha \bullet (DMT(t) + \rho) \qquad (5)$$

which effectively adds additional control to the system as a function of increasing DMT, in a manner that is regionally skewed by 2a density. In order to estimate the scaling parameter α, we performed a grid search over values [1, 10, 20, 30, 40, 50, 60, 70] and chose the value which minimized the Euclidean distance between the simulated output and the empirical DMT control energy on a group-level (SI Fig. 13; alpha = 30). DMT concentration over time was simulated from previously published population-level pharmacokinetic parameter estimates to obtain the typical predicted concentrations after a bolus dose of 20 mg DMT fumarate (using the R package *mrgsolve*)[60]. Specifically, theoretical effect compartment concentrations (based on EEG Alpha rhythms) were operationalized as $DMT(t)$. While plasma concentrations are based on observed/measured concentrations from collected blood samples, effect compartment concentrations are theoretical concentrations based on observed/measured effect data (in this case, alpha waves). The effect compartment is a way to link plasma concentrations to effect when it is believed that the effects are driven by concentrations in the biophase rather than plasma itself. This can be the case when the time course of the effects is slightly delayed as compared to plasma concentrations, explained by the fact that it takes a certain time for the drug to reach the effect compartment[98]. The regional serotonin 2a receptor density ρ was derived from previously published PET data (see "5-HT receptor mapping" section)[39]. In SI Fig. 13, we repeated our simulation in two ways, using (a) DMT plasma concentration estimates as $DMT(t)$, and (b) using a vector for ρ that contains the same total sum as the one constructed of serotonin 2a receptor density, but uniformly spread across regions. By having better correspondence with the empirical DMT time-series, these alternative simulations demonstrate the advantages of using brain-effect compartment concentrations over plasma concentrations, and skewing control by the serotonin 2a receptor distribution.

## Statistics and reproducibility

All pairwise comparisons between DMT and placebo time-series were made using cluster-based permutation testing[88] with a significance threshold of α = 0.05. All correlation coefficients reported were obtained from Spearman rank correlations, and p-values were obtained from comparison against either 10,000 random or spin permutations, depending on context. P-values were corrected for multiple comparisons using the Benjamini–Hochberg method. For within-subject (between-condition) direct comparisons, n = 14 participants. For correlational analyses over time, n = 838 time-points for correlations involving EEG and 28 time-points for correlations involving subjective intensity ratings. For brain correlations, n = 100 cortical or 116 cortical and subcortical regions. Analyses were produced with and without global signal regression.

## Reporting summary

Further information on research design is available in the Nature Portfolio Reporting Summary linked to this article.

## Data availability
Regional fMRI time-series, LZ, structural connectome, regional receptor densities, and intensity ratings are published at the code repository: https://github.com/singlesp/DMT_NCT and on Zenodo[99]. The source data behind the graphs in the paper can be found in Supplementary Data 1. All other data are available upon request.

## Code availability
Code to reproduce this analysis is available at: https://github.com/singlesp/DMT_NCT and on Zenodo[99]. The analysis was carried out using MATLAB R2020a Update 3 (9.8.0.1396136).

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

## Acknowledgements
This work was funded in part by NIH grants: R01NS102646 (AK) and RF1MH123232 (AK). AIL is supported by the Molson Neuro-Engineering Fellowship and FRQNT Strategic Clusters Program (2020-RS4-265502 - Centre UNIQUE - Union Neuroscience & Artificial Intelligence - Quebec) via the UNIQUE Neuro-AI Excellence Scholarship. The original DMT EEG-fMRI study data collection was made possible by donations from Patrick Vernon, mediated by the Beckley Foundation, as well as supplementary support from Anton Bilton and other founders of the Centre for Psychedelic Research, Imperial College London. CT is supported by funders for the Centre for Psychedelic Research at Imperial College London. RLC-H is supported by the Ralph Metzner Chair in Neurology and Psychiatry at UCSF.

## Author contributions
SPS conceptualized and performed the analysis and wrote the initial draft of the manuscript. CT designed the original study and collected the EEG-fMRI data, performed LZ calculations, and additionally provided early feedback on the analysis, figures, and manuscript. AIL provided early feedback on the analysis, figures, and manuscript. EE developed the PKPD models used for simulating DMT's impacts on control energy. LR preprocessed the EEG-fMRI data. RLC-H acquired funding for and supervised the original EEG-fMRI study. AK supervised the present analysis. All authors read and provided feedback on the manuscript.

## Competing interests
RLC-H is a scientific advisor to TRYP Therapeutics, Usona Institute, Journey Collab, Osmind, Maya Health, Beckley Psytech, Anuma, MindState, and Entheos Labs. All other authors have no conflicts of interest to declare.
