## [Transparent Peer Review file · Communications Biology]

Network control energy reductions under DMT relate to serotonin receptors, signal diversity, and subjective experience

Corresponding Author: Dr S. Singleton

Version 0:

Reviewer comments:

Reviewer #1

(Remarks to the Author)

This study investigates the effects of DMT, a serotonergic psychedelic substance, on brain activity and dynamics. Using a combination of EEG and fMRI recordings, it finds that DMT significantly reduces the control energy required for transitions between different brain states compared to a placebo. These changes are associated with increased EEG signal diversity and subjective drug intensity ratings. Furthermore, the study demonstrates that DMT's impact on brain dynamics can be simulated using a pharmacologically-informed control framework, shedding light on the mechanisms behind DMT's effects.

This study explores the impact of DMT on brain dynamics through a time-resolved network control analysis, a relatively novel method in psychedelic research. It integrates EEG and fMRI data, benefiting from both high spatial and temporal resolution. Furthermore, the modeling results strengthen the reliability of the findings. However, for greater reliability, a clearer explanation of the results and additional analysis is recommended. Moreover, discussing the similarities and differences in neural mechanisms compared to other psychedelic substances would be valuable.

1. In EEG analysis, why was Lempel-Ziv complexity chosen over other commonly used methods and measures to evaluate EEG signal complexity or information content?
2. In Figure 2b, it seems that EEG diversity has not undergone statistical analysis. To enhance clarity and facilitate comparison, it would be beneficial to directly present EEG diversity values for both the DMT and PLB conditions, following a format similar to Figure 2a.
3. Figures 2b and 2c include the pre-injection period in the correlation analysis. Could you provide the results when considering only the 20-minute post-administration period?
4. The diversity metric was computed using EEG data. It may be advantageous to also calculate diversity based on fMRI data to improve comparability with prior fMRI dynamic studies.
5. In Figure 3a, the left figure illustrates the change in regional control energy from the DMT pre-injection to post-injection condition, rather than comparing the DMT and PLB conditions as shown in Figure 2. Clarification is needed regarding the variation in baseline compared to Figure 2. Additionally, the rationale for using an 8-minute post-injection condition in this analysis, instead of the 20-minute period, should be provided for clarity.
6. In Figure 3a, the middle and right figures show correlation maps calculated with post-CE instead of Δ CE. Could you explain the reason for this difference in the index used?
7. Multiple comparisons correction should be applied, for example, in the results of Figure 3, Figure 2, SI Figure 3, SI Figure 4.
8. In Supplementary Figure 3d, the middle figure lacks r and p-values.

9. Psychedelic neuroimaging studies often encounter challenges related to significant motion artifacts. To strengthen the argument that the results remain unaffected by such artifacts, it would be beneficial to consider incorporating the frame-wise displacement (FD) covariate into the main correlation analysis in the supplemental analyses.
10. What specifically does the term 'subjective drug intensity' refer to, and what were the instructions provided for the rating?
11. It would be valuable to discuss the similarities and differences in neural mechanisms compared to other psychedelic substances.
12. Although the fMRI data offers high spatial resolution, the discussion section lacks sufficient explanation of the spatial results. It's evident that some brain regions and networks have a closer relationship with DMT. Enhancing the discussion with functional interpretations of these specific brain regions would be valuable.

Reviewer #2

(Remarks to the Author)

This is a super interesting study on the temporal and spatial effects of DMT on control energy and its relation to brain entropy and drug experience intensity implementing a novel combination of various advanced approaches including simultaneous fMRI/EEG, Lempel-Ziv complexity, dominance analysis, and network control theory. It is also a very important study because it verifies recent findings investigating the impact of psychedelics on brain dynamics and network transitions using different methodological approaches and different psychoplastogens. The manuscript also gives a clear introduction to the topic. Considering the information given for the methods (which are somewhat limited), the study seems very well conceptualized and implemented. I believe it is of significant interest to the community once published. However, I do have 2 main criticisms that need to be addressed before the work can be fully evaluated:

1. Most importantly, the description and justification of the methods are insufficient. Although the methods seem sound (except for some minor issues I raise below), I cannot evaluate them completely, because they are not thoroughly described nor justified. Even though the minimum of description is given compared to most other publications in the field, in my opinion this is just not enough. It is extremely important that the field starts to become transparent regarding their choices and arguments and potential lack of knowledge (we all know that there does not exist one best solution for all processing steps and a lot of choices are made randomly because we lack the empirical evidence). But if no paper is open about the reasons for their choices, a lot of valuable information is lost and the field will not improve. Also, I cannot verify completely that all methods are implemented accurately if they are not described and discussed in detail.
2. The structure of the manuscript in the results and methods section is very confusing which is partly due to the unfortunate structure of the journal to report the results before the methods. Nevertheless, information on the methods is broadly distributed throughout the manuscript; in the introduction, in the results, in the methods, and even in the figures. It is very difficult to retrieve all the necessary information, and thus, to understand what is going on exactly.

Detailed comments line-by-line:

1. Line 72-74: The authors speak of integrity here but do not really define what they mean by integrity, rather they cite different studies implementing different methods/measures. I believe it would be good for the field if these measures are more clearly explained/defined.
2. Line 99-106: I think these two sentences are key to this manuscript and would profit from a bit more explanation so that the reader, in case not so deep into the field of brain entropy and network control theory, can easily follow. Thus, it would be great if the authors could elaborate a bit more on what they mean by reduced control energy, enhanced state-transitioning effect, and how this reduction and enhancement is related to increased brain entropy. I am aware that this has been described in more detail in previous publications but it would be of value for the reader if this would be shortly summarized again and then explicitly referred to with something like "For a more detailed explanation/discussion see e.g., Singleton et al. 2022).
3. Lines 111-127: What I am missing here, instead of a description of the methods (repeated and extended in Figure 1 and other places throughout this manuscript), is a clear statement of the research questions that are addressed with this study.
4. Line 114: It says 8 minutes before and 20 minutes after injection but in lines 116-117 the authors write "These multimodal and continuous scanning conditions enable high temporal (EEG) and spatial (fMRI) resolution of brain activity before, during, and after an injection of DMT". It is unclear whether there actually exists data during injection. This information becomes only clear when you read the text of Figure 1a in which it says "a 28-minute long". Perhaps the authors want to provide this information earlier by writing instead: "undergoing 28 minutes of simultaneous electroencephalography (EEG) and fMRI recordings for 8 minutes before and 20 minutes after an intravenous (I.V.) bolus injection of DMT".
5. Lines 133-134: To better understand the results, it is essential to understand the study design. However, due to the fact that this journal unfortunately places the results and discussion before the methods section, the design is not 100% clear at this point. To make it easier to understand, perhaps the authors want to consider to revise this to something like "scanned twice per day on two different days (two-weeks apart) receiving either DMT or saline placebo at each of these separate days in a single-blind, counterbalanced design". Perhaps it would be helpful, instead of displaying the MRI icon, to present a

figure of the within-subject design showing details of the design, the number of scans in total and the time between each session.

6. Results section: Due to the structure of the journal, it is a bit difficult for the reader to understand the quite complex underlying methods when reading the results section. The manuscript scatters the information on methods across the results section, figures, and the methods section which makes it confusing. To improve readability, I would recommend to structure the methods section accordingly to the results section (by including headers for each analysis in the same order in which it is reported in the results section) and explicitly refer to the methods when reporting the results. See comments below.

7. Lines 154-157: The summary of the design at the beginning of the results section is similar confusing. Here it sounds as if there were only two sessions and each of this session was on a separate day ("acquired across two sessions, each on separate days"), which would result in a total of 2 measurements only (one placebo and one DMT). But if I understood correctly, there were 4 measurements in total; 2 sessions for placebo and additional 2 sessions for DMT.

8. Lines 161-162: "Control energy here is defined as the amount of input needed to drive the system from the current brain activity pattern to the next" – I think it would improve the understanding of the reader if this information would be provided much earlier when the term control energy is introduced for the first time.

9. Lines 161-164: Maybe it makes it easier for the reader if the authors would add a reference to the specific section in the methods section – something like: "see 'Minimum control energy' and 'Comparison of global control energy between DMT and PCB' in the methods section."

10. Lines 174-175: If I have not missed it, this is the only incident in which the authors report correction for multiple comparison (btw, this information belongs in the method section). However, it is unclear for how many comparisons they have corrected for and why they have only corrected the p-values from the two-sided, paired t-test. Please state this clearly in the methods section.

11. Lines 188-189: Please switch order of "subjective drug intensity ratings" and "signal diversity from simultaneous EEG" to match the order of results reported in this section.

12. Lines 191-192: Please explain in the methods section why differences between DMT and PCB were used to calculate correlation for global control energy (compared to pre- versus post for regional control energy)? It would make it easier for the reader to follow if choices would be explained.

13. Line 192: Please add a reference after "between-condition differences in these metrics" to the specific section in the methods section – something like: "see 'Correlation of global control energy with signal diversity and subjective drug intensity ratings' in the methods section."

14. Lines 194-198: If I understand correctly, the measures of global control energy were calculated from the first session while the ratings were obtained in the second session. I believe the authors should discuss in the methods section the variability of the subjective intensity ratings across sessions including the bias of possible carry-over effects, especially since participants would possibly be exposed to this drug for the first time in the first session compared to the second session which takes places just a bit later. I would also like to see a figure showing these results at the single-subject level to verify whether the pattern is similar across subjects (e.g., in the supplementary materials).

15. Lines 194-197: This information belongs into the methods section and could be reported in a much shorter version here including a reference to the specific section in the methods section.

16. Lines 211-212: Why did the authors choose to use difference between DMT and PCB for the global measure but post- vs pre-injection of DMT for the regional measure? Please justify in the methods section.

17. Lines 211-214: I really like how this is explained here. However, I think it would be better if all this information would be systematically gathered in the methods section.

18. Line 214: Please add a reference to the specific section in the methods section- something like: "see 'Correlation of spatial patterns of control energy with signal diversity and subjective drug intensity ratings' in the methods section. "

19. Lines 218-223: This part also belongs into the methods section.

20. Line 247: For clarity, it should read "with serotonin (5-HT) 2a receptor" to align with the phrasing below.

21. Line 252: Please add a reference to "Dominance analysis" in the methods section.

22. Line 274: Please add a reference to the specific section in the methods section- something like: "see 'Simulation from pharmacokinetics and the serotonin 2a receptor maps' in the methods section."

23. Line 279-296: These supplemental analyses also need to be included in the methods section.

24. Lines 279-280: Please discuss in the methods why it is important to look at the time-resolved control energies for a priori

resting-state networks.

25. Lines 285-286: Please state a reason in the methods why the main results need to be reproduced without global regression.

26. Lines 315-323: I really appreciate the short summary.

27. Line 361: Can the authors please clarify for the reader what they mean by “above and beyond the effect of spatial autocorrelation”

28. Line 363-364: The line of reasoning here may not be 100% clear. Not everyone may be completely familiar with the pharmacokinetics and -dynamics of psychedelics. Can the authors perhaps explain their choice of variables for the dominance analysis.

29. Lines 396-399: It is not clear to me why the authors are referring to SI Figure 4 here? I think they mean SI Figure 5?

30. Line 403: It would be great if the authors could include the literature references here.

31. Line 421: Can the authors add “...this is the third independent dataset including two previous published ones using LSD and psilocybin (Singleton et al. 2022) which has found...” to clarify this point?

32. Lines 447-454: Can the authors include information on the participants in the supplementary materials including inclusion criteria. Specifically, it is important to understand if and why subjects were excluded regarding previous drug experiences.

33. Methods section: The methods section is unfortunately not detailed enough to be able to completely comprehend what has been done without going through the provided code (and let's be honest here: which no reader does unless they are using it for their own study). Especially problematic is the fact that decisions are not justified including stating the lack of empirical evidence if applicable. I believe this is very important to make the field more transparent and guide decisions in future studies. In addition, the methods section is not well structured. The description of the methods is randomly distributed across the results section, the figures, the methods section, and references to previous studies. Thus, it becomes very difficult to gather all necessary information to understand what has been implemented and why. If there is not enough space, please move the methods section to the supplementary materials.

34. Lines 456-458: Please state how much time was between the first and the second session on each testing day.

35. Line 462: Please add duration of administration. You can guess from Figure 1 but, unless I have not missed it, it is nowhere explicitly stated. Also, I think it should read “of the 8th minute”.

36. Lines 464-466: Can the authors please clarify how the rating was implemented exactly.

37. Lines 466-467: Why did the authors choose a single-blind design? Please discuss and justify why blinding was chosen since one can criticize the futility of blinding subjects from their psychedelic experience due to ethical and pragmatical considerations as discussed here [Schenberg, E.E., Who is blind in psychedelic research? Letter to the editor regarding: blinding and expectancy confounds in psychedelic randomized controlled trials. *Expert Rev Clin Pharmacol*, 2021. 14(10): p. 1317-1319.]. Benefits potentially gained by blinding, though again, unlikely in research with compounds that induce psychedelic states, could instead be acquired via assessment of expectations for the psychedelic experience and incorporated into analyses using tools validated for this purpose such as the Credibility/Expectancy Questionnaire [DeVilly, G.J. and T.D. Borkovec, Psychometric properties of the credibility/expectancy questionnaire. *J Behav Ther Exp Psychiatry*, 2000. 31(2): p. 73-86.]. Have these assessments been considered?

38. Lines 469-470: Here the authors should maybe include a discussion on the variability of the subjective intensity ratings across sessions to the other including the discussion of possible carry-over effects – see comment 14.

39. Line 474: I generally do not find it appropriate if description of any part of the methods are provided only in a reference to which the reader may or may not have access. All relevant information should be provided in the paper at hand.

40. Lines 490-509: Same criticism as above. This is just not enough information (even when considering the necessity to read additional two more papers while trying to figure out which information is specific to the data analyzed in the present study). I would like to see a justification for the specific preprocessing choices (e.g., why did the authors choose AFNI (3dvolreg) for motion correction? Why did the authors choose to register each volume to the most similar volume in a least square sense and not to the first?). It is extremely important that the field starts to become transparent.

41. Lines 499-500: Why did the authors choose to use scrubbing and nuisance regression as a denoising strategy method? Please discuss (see e.g., Parkes et al., 2018, *Neuroimage*). How exactly did the authors regress out 9 nuisance regressors? What are the 3 anatomically-related regressors? How were they retrieved?

42. Line 510: I am missing the details regarding EEG preprocessing. I am no expert on EEG but I am quite sure that you do need to preprocess the data before you can run Lempel Ziv.

43. Line 522-525: Please discuss why the same parameters as in prior work were chosen.

44. Lines 539-540: Again, I find it too much to ask for the reader to look up multiple different papers and figure out which information details fit the data reported in this manuscript. Please provide this information in the present publication. If word restriction is an issue, put the methods into the supplementary materials.

45. Lines 544-551: Can the authors please add a short discussion why Lempel-Ziv was chosen as the measure of signal diversity compared to other approaches measuring entropy that have been introduced lately to detect the level of complexity in the brain as a guide for future implementation.

46. Line 552: Maybe it is better to put the section "Minimum control energy" here to follow the structure of the reported results and make it easier for the reader. Also add a section describing the stats for comparing global control energy between DMT and PCB as described in Figure 2a for which the authors present the results in the results section "Global control energy is lower after DMT infusion versus after placebo". Then, add a section describing and briefly justifying the Pearson correlation implemented to obtain the results in section "Global control energy under DMT negatively correlates with subjective drug intensity ratings and signal diversity from simultaneous EEG" (it is not clear what the authors have done exactly – permutation?). It would be good if each section has its own header following the structure of the results – so the authors can refer to the specific method section when reporting the results.

47. Lines 553-567: This part is very clear and well explained and also justified.

48. Lines 569-624: This part should move up to match the order in the results section (see comment above).

49. Lines 585-593: If I understand correctly, this part here concerns only the DMT simulation. To make this easier to relate to the results, I would move this part into its own section to match the structure of the results section. Also, this part is not 100% clear. As it is argued now, I do not understand what kind of measures were exactly taking to "demonstrate the advantage of incorporating both the simulated brain-effect concentration (compared to simulated plasma concentration)" (lines 292-293). In Figure SI 5, results comparing these two approaches are shown. How was the simulated plasma concentration calculated and how does it differ from simulated brain-effect concentration (defined as "independently validated model of DMT's pharmacokinetic impact on EEG alpha rhythms to simulate population-level 'brain-effect' of DMT over the course of the 28 minute scans (Eckernäs et al. 2023)") to show this advantage?

50. Line 596: Should this not read SI Figure 5?

51. SI Figure 4: The @ as symbol for alpha is confusing.

Reviewer #3

(Remarks to the Author)

This is a fascinating manuscript that would be a valuable contribution if published as is. However, there are a few issues that may be valuable to address.

Did the authors consider using the average, modal, and boundary controllability analyses described here? If not, why not?

Perhaps a brief comment about further work in that direction could be helpful.

<https://www.nature.com/articles/ncomms9414/>

The description of dominance analyses was very interesting/compelling. Was this the first analysis technique used? If another method was used and found to be lacking, perhaps it could be included as a supplementary material? If this was the only technique used, a little more up front description of its a priori justification could be helpful for some potentially skeptical readers (although I am personally compelled).

"In recent work using a receptor-informed network control theory framework, we demonstrated that the serotonergic psychedelics lysergic acid diethylamide (LSD) and psilocybin flatten the brain's control energy landscape in a manner that covaries with more dynamic and entropic brain activity."

In this previous work, more dynamic/entropic brain activity was interpreted as indicating a flattened energy landscape, in support of the REBUS model. However, an "ALBUS" model provides an alternative (Entropic Brain Theory compatible) interpretation in which these more diverse states are reached via more powerful/persistent predictions, potentially especially those associated with intermediate-level conscious representations:

<https://psyarxiv.com/zqh4b/>

That is, rather than (or in addition to) flattening the terrain, psychedelics might also help with climbing peaks (and perhaps escaping from local optima). Do the methods used in this paper help to disambiguate these possibilities, or is this still an open question?

Version 1:

Reviewer comments:

Reviewer #1

(Remarks to the Author)

My questions have been fully addressed by the authors. I believe the additional analyses and the more comprehensive and in-depth discussion have significantly improved the manuscript. I particularly appreciate that this EEG-fMRI dataset of DMT captures the entire administration process, allowing for a more comprehensive understanding of how DMT dynamically influences brain activity over time.

Although the revision process took a long time, I believe the wait was entirely worth it. I appreciate the authors' effort in refining and enhancing this work.

Minor Suggestions:

- The figure legends in the supplementary materials could be more detailed. For example, clarifying what analyses were performed in each figure, what specific indices represent, and the definitions of abbreviations would improve clarity.
- In Figures 7 and 8, some plots are not fully displayed, and certain data points appear to be cut off.

Reviewer #2

(Remarks to the Author)

The authors have answered all the questions and improved the manuscript fundamentally. With the additional information, there is one issue that still needs to be addressed:

1. Global control energy analyses: More information is needed for how the normalization procedures were implemented to calculate control energy. At least provide references.

Dear reviewers,

We would like to thank each of you for the thoughtful comments and thorough evaluation of our manuscript. Our manuscript has greatly improved from responding to your feedback. We have provided several sensitivity analyses and additional reporting and clarifications on our methods. We would also like to issue a sincere apology for the timing of our resubmission, which was delayed due to several significant changes to the life of the first-author since the original submission.

Reviewers' comments:

Reviewer #1 (Remarks to the Author):

This study investigates the effects of DMT, a serotonergic psychedelic substance, on brain activity and dynamics. Using a combination of EEG and fMRI recordings, it finds that DMT significantly reduces the control energy required for transitions between different brain states compared to a placebo. These changes are associated with increased EEG signal diversity and subjective drug intensity ratings. Furthermore, the study demonstrates that DMT's impact on brain dynamics can be simulated using a pharmacologically-informed control framework, shedding light on the mechanisms behind DMT's effects.

This study explores the impact of DMT on brain dynamics through a time-resolved network control analysis, a relatively novel method in psychedelic research. It integrates EEG and fMRI data, benefiting from both high spatial and temporal resolution. Furthermore, the modeling results strengthen the reliability of the findings. However, for greater reliability, a clearer explanation of the results and additional analysis is recommended. Moreover, discussing the similarities and differences in neural mechanisms compared to other psychedelic substances would be valuable.

1. In EEG analysis, why was Lempel-Ziv complexity chosen over other commonly used methods and measures to evaluate EEG signal complexity or information content?

Lempel-Ziv complexity of EEG/MEG is a broadly replicated signature of psychedelic states ([10.1371/journal.pone.0242056](https://doi.org/10.1371/journal.pone.0242056), [10.1016/j.neuroimage.2019.03.076](https://doi.org/10.1016/j.neuroimage.2019.03.076), [10.1038/srep46421](https://doi.org/10.1038/srep46421)), including with DMT ([10.1038/s41598-019-51974-4](https://doi.org/10.1038/s41598-019-51974-4), [10.1002/psp4.12933](https://doi.org/10.1002/psp4.12933), [10.1177/0269881120981384](https://doi.org/10.1177/0269881120981384)). In addition, the parent study ([10.1073/pnas.2218949120](https://doi.org/10.1073/pnas.2218949120)) used this same method for their assessment of EEG signal complexity, and here we sought to understand how control energy relates to the previous findings rather than to introduce an entirely new assessment of EEG signal diversity in this dataset.

2. In Figure 2b, it seems that EEG diversity has not undergone statistical analysis. To enhance clarity and facilitate comparison, it would be beneficial to directly present EEG diversity values for both the DMT and PLB conditions, following a format similar to Figure 2a.

Thank you for this suggestion. We have added the following SI Figure.

SI Figure 3: LZ_c comparison. Gray boxes reflect cluster-corrected significant time-points (see *EEG preprocessing and signal diversity calculation* for details).

3. Figures 2b and 2c include the pre-injection period in the correlation analysis. Could you provide the results when considering only the 20-minute post-administration period?

We have run these calculations and now include them in-text:

“Recalculating each correlation using only the 20-minute post-administration period reveals a significant negative correlation between control energy and signal diversity (Spearman’s $R = -0.113$, $p_{perm} = 0.0053$) and a non-significant correlation between control energy and intensity (Spearman’s $R = 0.2827$, $p_{perm} = 0.8897$). The lack of correlation in the latter case is likely due to the fact that because intensity ratings were collected at the end of each minute (i.e. at the end of the last minute prior to DMT injection, and then again one minute after injection), 1) there are fewer sampled points on which to capture the relationship, so it will be noisier, and 2) the window of the largest change in intensity and control energy (right after administration) is not well sampled.”

4. The diversity metric was computed using EEG data. It may be advantageous to also calculate diversity based on fMRI data to improve comparability with prior fMRI dynamic studies.

The advantage of using EEG data was that we could obtain a diversity metric corresponding to each fMRI time-point, allowing us to study the relationship between

time-resolved control energy based on fMRI and neural diversity based on EEG over the duration of the rapidly evolving DMT experience. fMRI, which measures oxygenated blood flow, may be impacted by the vascular effects of psychedelics whereas EEG directly measures neural activity. Having a non-vascular measure of neural activity with which to compare fMRI based findings is one of the strengths of this study. Additionally, relying on the fMRI data for diversity metrics would limit us to exploring relationships between averaged control energy and diversity across individuals, a phenomenon that has been explored previously. We've added the following to the discussion:

“In our own recent work using an approach similar to the one employed here, we found that LSD and psilocybin decreased control energy, and, across individuals, larger decreases under LSD were associated with more complex brain-state sequences extracted from fMRI (Singleton et al. 2022). Other recent studies have also found associations across individuals between control energy and fMRI-based entropy or diversity metrics (Tozlu et al. 2023; Singleton et al. 2024). The present study design enables us to further strengthen the association between control energy and neural entropy by showing that the global control energy throughout the fMRI time-series is temporally coupled with neural signal diversity measured with simultaneous EEG, the latter of which does not depend on neurovascular coupling which are likely altered under psychedelics (Figure 2b).”

5. In Figure 3a, the left figure illustrates the change in regional control energy from the DMT pre-injection to post-injection condition, rather than comparing the DMT and PLB conditions as shown in Figure 2. Clarification is needed regarding the variation in baseline compared to Figure 2. Additionally, the rationale for using an 8-minute post-injection condition in this analysis, instead of the 20-minute period, should be provided for clarity.

We have added the following passages to the text to clarify these decisions (below). Also, see the response to comment 6.

In text:

Next, we interrogate control energy specifically under DMT, taking advantage of our in-scanner drug-delivery design to accurately measure (and later simulate) DMT's effects in real-time. We start by investigating control energy under DMT at the regional, rather than global, level in order to understand spatial relationships.

New Methods section:

Regional control energy analyses

Having demonstrated DMT's impact on control energy with respect to placebo, and its relationship to EEG signal diversity and subjective intensity ratings, we next sought to understand the regional relationships of control energy specifically under DMT and its relationship to the serotonin 2a receptor's spatial distribution. First, we calculated each region's mean change in control energy during the first 8 minutes post-injection compared to and adjusted by the region's mean control energy during the 8 minute pre-injection window to

summarize DMT's peak effects on regional control energy and to ensure the same amount of data was going into both pre and post injection metrics (Figure 3a, left). Next, we correlated each region's mean (over subjects) control energy time-series under DMT (full 28-minute scan) with EEG signal complexity (Figure 3a, middle), and lastly we correlated each region's mean control energy time-series (windowed over 1-minute intervals) with subjective intensity ratings (Figure 3a, right). Each of these metrics were then used as regional spatial maps to compare against the serotonin 2a receptor (see *5-HT receptor mapping* for acquisition details). The cortical values of each metric was Spearman-rank correlated with the serotonin 2a spatial distribution and p-values were generated via comparison against 10,000 spin permutations (Váša et al. 2018) to test for significance above-and-beyond the effects of spatial auto-correlation in the data (Figure 3c) and then subsequently corrected for multiple comparisons (Benjamini and Hochberg 1995). These correlations were repeated including the subcortical values (SI Fig. XX).

6. In Figure 3a, the middle and right figures show correlation maps calculated with post-CE instead of Δ CE. Could you explain the reason for this difference in the index used?

Following Figure 2, which presents a coarse-grained overview of the temporal differences between the DMT and the placebo conditions, and their relationships with diversity and intensity over time, we desired to specifically interrogate the the DMT condition for the remainder of the analysis, taking advantage of the in-scanner drug delivery design to accurately measure and simulate DMT's effects in real time. Figures 3 and 4 are meant to probe the DMT condition on its own, to justify and inform our later simulation. In Figure 3, the left metric interrogates the Δ CE of each region using averages over the pre-injection (minutes 1 to 8) and the first 8 minutes of the post-injection period (minutes 9 to 16). We used the first 8 minutes of the post-injection period as it is the time-period where DMT's effects are the greatest and is the same amount of data as the pre-injection period. The middle and right plots each use regional correlations of CE over the full DMT scans. We have added text clarifying and justifying these choices (above). However, we can see how one may be interested in additional analyses that were not performed. We have added the following figures to the SI, performing the same interrogation of the placebo condition as a negative control:

SI Figure 4: Condition-specific correlations. (a) Group-level correlation between CE and LZ_c for the DMT condition ($p_{\text{perm}} < 0.0001$). (b) Group-level correlation between CE and intensity for the DMT condition ($p_{\text{perm}} < 0.0001$). (c) Group-level correlation between CE and LZ_c for the placebo condition ($p_{\text{perm}} < 0.0001$). (d) Group-level correlation between CE and intensity for the placebo condition ($p_{\text{perm}} = 0.0097$).

SI Figure 9: Replication of analyses from Main Text Fig. 3 using the placebo scans instead of DMT scans.

SI Figure 10: Replication of analyses from Main Text Fig. 4 using the placebo scans instead of DMT scans.

7. Multiple comparisons correction should be applied, for example, in the results of Figure 3, Figure 2, SI Figure 3, SI Figure 4.

All figures/captions/and in-text p-values have been updated to show/indicate corrected p-values.

8. In Supplementary Figure 3d, the middle figure lacks r and p-values.

Thank you for catching this. We have updated the figure.

9. Psychedelic neuroimaging studies often encounter challenges related to significant motion artifacts. To strengthen the argument that the results remain unaffected by such artifacts, it

would be beneficial to consider incorporating the frame-wise displacement (FD) covariate into the main correlation analysis in the supplemental analyses.

Thank you for this suggestion. We have run partial correlations with the TR by TR framewise displacement in the main correlations from Figure 2, which we now include in-text:

“ We additionally recalculate the main correlations from Figure 2 using group-averaged framewise displacement values as a covariate of non-interest (Spearman’s $R = -0.375$, $p_{\text{perm}} < 0.0001$ and Spearman’s $R = -0.516$, $p_{\text{perm}} = 0.0015$, respectively).”

10. What specifically does the term 'subjective drug intensity' refer to, and what were the instructions provided for the rating?

Subjective drug intensity refers to how intense participants rate drug effects on a scale of 0-10. Specifically, the ratings were collected by playing an audio cue (“Intensity from 0 to 10”) through MR-compatible earphones which was followed by the participants’ verbal response collected via an MR-compatible microphone. We have updated the methods to reflect this.

11. It would be valuable to discuss the similarities and differences in neural mechanisms compared to other psychedelic substances.

Thank you for this suggestion - we have added the following to the discussion:

A recent dual-receptor model of psychedelic action positions the serotonin 1a receptor as an influential modulator of their effects (Juliani et al 2024). Compared with other classic psychedelics like LSD and psilocybin, DMT has a higher serotonin 2a:1a receptor binding ratio (Rickli et al 2016). While, as discussed above, the serotonin 2a receptor is the primary receptor responsible for much of psychedelics’ neural and subjective effects, their action at other serotonin receptors is of increasing interest to the field. Indeed rodent models have suggested that some of psychedelics’ antidepressant and synaptogenic effects may act through non-2a mechanisms (Hesselgrave et al 2021, Shao et al 2021). Among other serotonin receptors considered here after the 2a receptor, DMT’s effects were best explained by the serotonin 1b receptor, closely followed by the 1a receptor. This finding is consistent with our previous work in LSD and psilocybin (Singleton et al 2022).

12. Although the fMRI data offers high spatial resolution, the discussion section lacks sufficient explanation of the spatial results. It’s evident that some brain regions and networks have a closer relationship with DMT. Enhancing the discussion with functional interpretations of these specific brain regions would be valuable.

Thank you for this comment, we have added a paragraph on the network findings to the discussion:

“On a network level, DMT exerted its effects most strongly on the visual, frontoparietal, and default mode networks (SI Figure 1). DMT, and other psychedelics more broadly, are known to preferentially impact all three of these networks (Shinozuka et al 2024), albeit findings in the visual network have been less consistent. Our time-resolved design may shed some light on this inconsistency. While DMT’s effects on the control energy of the frontoparietal and default mode networks are strongest during the first-half of the post-injection period, the opposite is true for the visual network (SI Fig. 2). Notably, DMT’s large effects in the visual network in the second-half of the post-injection scanning period appear to be due to control energy gradually increasing throughout the post-injection scanning period in the placebo condition, while low-levels are maintained in the DMT condition throughout. This may in-part be an arousal effect. BOLD amplitude is known to fluctuate with the degree of wakefulness (Liu and Falahpour 2020), and BOLD amplitude in the visual cortex in particular increases as arousal drops and individuals transition to light sleep (Fukunaga et al 2006). We speculate that arousal under the placebo and DMT conditions is high around the injection period, and remains high in the DMT conditions but slowly decreases over time in the placebo condition, leading to increases in control energy for the visual network under placebo and a maintained reduction in control energy under DMT.”

Reviewer #2 (Remarks to the Author):

This is a super interesting study on the temporal and spatial effects of DMT on control energy and its relation to brain entropy and drug experience intensity implementing a novel combination of various advanced approaches including simultaneous fMRI/EEG, Lempel-Ziv complexity, dominance analysis, and network control theory. It is also a very important study because it verifies recent findings investigating the impact of psychedelics on brain dynamics and network transitions using different methodological approaches and different psychoplastogens. The manuscript also gives a clear introduction to the topic. Considering the information given for the methods (which are somewhat limited), the study seems very well conceptualized and implemented. I believe it is of significant interest to the community once published. However, I do have 2 main criticisms that need to be addressed before the work can be fully evaluated:

1. Most importantly, the description and justification of the methods are insufficient. Although the methods seem sound (except for some minor issues I raise below), I cannot evaluate them completely, because they are not thoroughly described nor justified. Even though the minimum of description is given compared to most other publications in the field, in my opinion this is just not enough. It is extremely important that the field starts to become transparent regarding their choices and arguments and potential lack of knowledge (we all know that there does not exist

one best solution for all processing steps and a lot of choices are made randomly because we lack the empirical evidence). But if no paper is open about the reasons for their choices, a lot of valuable information is lost and the field will not improve. Also, I cannot verify completely that all methods are implemented accurately if they are not described and discussed in detail.

2. The structure of the manuscript in the results and methods section is very confusing which is partly due to the unfortunate structure of the journal to report the results before the methods. Nevertheless, information on the methods is broadly distributed throughout the manuscript; in the introduction, in the results, in the methods, and even in the figures. It is very difficult to retrieve all the necessary information, and thus, to understand what is going on exactly.

Detailed comments line-by-line:

1. Line 72-74: The authors speak of integrity here but do not really define what they mean by integrity, rather they cite different studies implementing different methods/measures. I believe it would be good for the field if these measures are more clearly explained/defined.

We have replaced 'integrity' here with 'functional connectivity' to be more clear.

2. Line 99-106: I think these two sentences are key to this manuscript and would profit from a bit more explanation so that the reader, in case not so deep into the field of brain entropy and network control theory, can easily follow. Thus, it would be great if the authors could elaborate a bit more on what they mean by reduced control energy, enhanced state-transitioning effect, and how this reduction and enhancement is related to increased brain entropy. I am aware that this has been described in more detail in previous publications but it would be of value for the reader if this would be shortly summarized again and then explicitly referred to with something like "For a more detailed explanation/discussion see e.g., Singleton et al. 2022).

We have edited/added the highlighted clarifying text:

"LSD and psilocybin reduces the control energy required to transition between task-free fMRI-derived brain-states in a manner that, across individuals, covaries with increases in brain activity entropy - i.e., the diversity or complexity of the brain's spontaneous oscillations recorded across time, a well-known marker of psychedelic action (Carhart-Harris 2018). Reduced control energy here indicates more facile state transitions under the network control framework, thereby potentially enabling access to more complex sequences of brain activity. Moreover, we provided evidence that the reduced control energy effect of psychedelics is associated with the brain's spatial distribution of the 5-HT_{2a} receptor expression (Singleton et al. 2022)."

3. Lines 111-127: What I am missing here, instead of a description of the methods (repeated and extended in Figure 1 and other places throughout this manuscript), is a clear statement of the research questions that are addressed with this study.

We have added the highlighted text to clarify our aims:

“In the present analysis our aims are twofold. First, we seek to quantify DMT’s effects on control energy, and their relation to the serotonin 2a receptor, neural entropy, and subjective effects. Given the rapid kinetics of DMT’s effects, the use of time-resolved analysis techniques will be crucial for capturing changes in the brain’s activation dynamics in real time. Therefore our second aim is to evolve our methods in order to address this challenge. Here, we employ a time-resolved network control analysis of...”

4. Line 114: It says 8 minutes before and 20 minutes after injection but in lines 116-117 the authors write “These multimodal and continuous scanning conditions enable high temporal (EEG) and spatial (fMRI) resolution of brain activity before, during, and after an injection of DMT”. It is unclear whether there actually exists data during injection. This information becomes only clear when you read the text of Figure 1a in which it says “a 28-minute long”. Perhaps the authors want to provide this information earlier by writing instead: “undergoing 28 minutes of simultaneous electroencephalography (EEG) and fMRI recordings for 8 minutes before and 20 minutes after an intravenous (I.V.) bolus injection of DMT”.

We have edited the text exactly as suggested.

5. Lines 133-134: To better understand the results, it is essential to understand the study design. However, due to the fact that this journal unfortunately places the results and discussion before the methods section, the design is not 100% clear at this point. To make it easier to understand, perhaps the authors want to consider to revise this to something like “scanned twice per day on two different days (two-weeks apart) receiving either DMT or saline placebo at each of these separate days in a single-blind, counterbalanced design”. Perhaps it would be helpful, instead of displaying the MRI icon, to present a figure of the within-subject design showing details of the design, the number of scans in total and the time between each session.

We have edited the text exactly as suggested.

6. Results section: Due to the structure of the journal, it is a bit difficult for the reader to understand the quite complex underlying methods when reading the results section. The manuscript scatters the information on methods across the results section, figures, and the methods section which makes it confusing. To improve readability, I would recommend to structure the methods section accordingly to the results section (by including headers for each analysis in the same order in which it is reported in the results section) and explicitly refer to the methods when reporting the results. See comments below.

We have significantly re-structured the methods section to comply with the above suggestion.

7. Lines 154-157: The summary of the design at the beginning of the results section is similar confusing. Here it sounds as if there were only two sessions and each of this session was on a

separate day (“acquired across two sessions, each on separate days”), which would result in a total of 2 measurements only (one placebo and one DMT). But if I understood correctly, there were 4 measurements in total; 2 sessions for placebo and additional 2 sessions for DMT.

We have edited this to more clearly represent the design, using ‘visits’ to refer to separate visits rather than ‘sessions’.

“We analyzed simultaneous EEG-fMRI resting-state data for 14 participants acquired across two visits (one drug visit, and one placebo visit), each on separate days (Timmermann et al. 2023). At each visit, two scanning sessions comprising 28 minute long resting-state EEG-fMRI scans were collected, with I.V. bolus infusion of either DMT or placebo at the end of the 8th minute (Figure 1a). The second scanning session included the collection of subjective drug-intensity ratings (0-10) at the end of every minute, while the first scanning session was resting-state. This report analyses the EEG-fMRI data of the resting-state scan, while using the intensity ratings of the second session for correlational analyses. See *Participants and study design* and *EEG-fMRI acquisition* for full details on study design and acquisition.”

8. Lines 161-162: “Control energy here is defined as the amount of input needed to drive the system from the current brain activity pattern to the next” – I think it would improve the understanding of the reader if this information would be provided much earlier when the term control energy is introduced for the first time.

Thank you for this suggestion, we have added this to the introduction where control energy is first introduced.

9. Lines 161-164: Maybe it makes it easier for the reader if the authors would add a reference to the specific section in the methods section – something like: “see ‘Minimum control energy’ and ‘Comparison of global control energy between DMT and PCB’ in the methods section.”

We have added references to the appropriate methods sections throughout.

10. Lines 174-175: If I have not missed it, this is the only incident in which the authors report correction for multiple comparison (btw, this information belongs in the method section). However, it is unclear for how many comparisons they have corrected for and why they have only corrected the p-values from the two-sided, paired t-test. Please state this clearly in the methods section.

We have updated all figures/captions to reflect corrected p-values and have added the appropriate information to the methods section.

11. Lines 188-189: Please switch order of “subjective drug intensity ratings” and “signal diversity from simultaneous EEG” to match the order of results reported in this section.

We have updated this text as requested.

12. Lines 191-192: Please explain in the methods section why differences between DMT and PCB were used to calculate correlation for global control energy (compared to pre- versus post for regional control energy)? It would make it easier for the reader to follow if choices would be explained.

In addition to adding additional related supplemental analyses (see our response to Reviewer 1's comment #6), we have added reasoning of these decisions to our new methods section *Global control energy analyses*.

13. Line 192: Please add a reference after “between-condition differences in these metrics” to the specific section in the methods section – something like: “see ‘Correlation of global control energy with signal diversity and subjective drug intensity ratings’ in the methods section.”

We have added reference to the appropriate methods section.

14. Lines 194-198: If I understand correctly, the measures of global control energy were calculated from the first session while the ratings were obtained in the second session. I believe the authors should discuss in the methods section the variability of the subjective intensity ratings across sessions including the bias of possible carry-over effects, especially since participants would possibly be exposed to this drug for the first time in the first session compared to the second session which takes places just a bit later. I would also like to see a figure showing these results at the single-subject level to verify whether the pattern is similar across subjects (e.g., in the supplementary materials).

We have modified our methods section, *Participants and study design*, to more clearly present the design and order of drug vs placebo administrations. All drug sessions were on separate testing days (separated by two weeks). Due to the differential tolerance of biological and subjective effects to DMT in humans (R. J. Strassman, Qualls, and Berg 1996), carryover effects from one testing day to the other are not expected, and the subjective intensity experienced for the same dose is expected to be roughly similar between testing days.

We have added subject level plots and correlations to the SI.

SI Figure 5: Individual-level correlations between CE and LZ_c for the DMT condition.

SI Figure 6: Individual-level correlations between CE and LZ_c for the placebo condition.

SI Figure 7: Individual-level correlations between CE and subjective intensity for the DMT condition.

SI Figure 8: Individual-level correlations between CE and subjective intensity for the placebo condition. Several individuals rated intensity at 0 for every time-point and correlations could not be computed for these subjects.

15. Lines 194-197: This information belongs into the methods section and could be reported in a much shorter version here including a reference to the specific section in the methods section.

We have shortened this description and refer the reader to the appropriate methods section.

16. Lines 211-212: Why did the authors choose to use difference between DMT and PCB for the global measure but post- vs pre-injection of DMT for the regional measure? Please justify in the methods section.

Please see our response to Reviewer 1. We have added a new methods section *Regional control energy analysis* that details this decision.

17. Lines 211-214: I really like how this is explained here. However, I think it would be better if all this information would be systematically gathered in the methods section.

We now refer the reader to the appropriate methods section (*Regional control energy analyses*).

18. Line 214: Please add a reference to the specific section in the methods section- something like: “see ‘Correlation of spatial patterns of control energy with signal diversity and subjective drug intensity ratings’ in the methods section. “

See above.

19. Lines 218-223: This part also belongs into the methods section.

We now include this information in the methods section.

20. Line 247: For clarity, it should read “with serotonin (5-HT) 2a receptor” to align with the phrasing below.

We have edited the text as suggested.

21. Line 252: Please add a reference to “Dominance analysis” in the methods section.

We now refer the reader to the appropriate methods section.

22. Line 274: Please add a reference to the specific section in the methods section- something like: “see ‘Simulation from pharmacokinetics and the serotonin 2a receptor maps’ in the methods section.”

We now refer the reader to the appropriate methods section.

23. Line 279-296: These supplemental analyses also need to be included in the methods section.

All supplemental analyses are now included in the methods section.

24. Lines 279-280: Please discuss in the methods why it is important to look at the time-resolved control energies for a priori resting-state networks.

We have updated the methods to address this. See the new *Network-level control energy analyses* section.

25. Lines 285-286: Please state a reason in the methods why the main results need to be reproduced without global regression.

We have added the following to the methods section:

“Because global signal regression remains a hotly debated topic, particularly in the context of altered states of consciousness (Tanabe et al. 2020; Mortaheb et al. 2024; Singleton and Kuceyeski 2024), we repeated our main analyses without the use of global signal regression (SI Fig. 11).”

26. Lines 315-323: I really appreciate the short summary.

Thank you!

27. Line 361: Can the authors please clarify for the reader what they mean by “above and beyond the effect of spatial autocorrelation”

We appreciate the request for clarification. In recent years, there has been increasing appreciation that brain maps exhibit spatial autocorrelation: in other words, whereas correlation assumes that data-points are independent, brain maps violate this assumption because two neighbouring regions will systematically exhibit values that are more similar, than two regions that are far apart. In other words, similarity of regional values is influenced by spatial proximity. In turn, this can result in inflated false positive rate: two random brain maps can still appear significantly spatially correlated, just as a result of spatial autocorrelation. Therefore, appropriate null models are required to demonstrate that when two brain maps of interest are spatially correlated, this spatial correlation is more than what would be expected from comparing random maps with similar levels of spatial autocorrelation (hence our phrase: “above and beyond the effect of spatial autocorrelation”). One popular method is to use so-called spin-nulls: projecting one of the maps on a sphere, then rotating the sphere randomly, and re-projecting the spun data onto the brain. The resulting spun map will have the same spatial autocorrelation as the original map, but allocated randomly in terms of neuroanatomy: therefore, its correlation with the map of interest will be purely due to spatial autocorrelation. By building a null distribution of such maps, one embodies the null hypothesis that correlation between maps of interest is only due to spatial autocorrelation. If the empirical correlation between maps is greater than what is observed in this null distribution, the null hypothesis that correlation is just a by-product of spatial autocorrelation can be rejected, indicating that anatomical location also plays a

role (see Alexander-Bloch et al., 2018 NeuroImage; Burt et al., 2020 NeuroImage; Markello & Music, 2021 NeuroImage; Vasa and Masic, 2022 Nature Reviews Neuroscience).

We have added much of the above text to the discussion for the reader.

28. Line 363-364: The line of reasoning here may not be 100% clear. Not everyone may be completely familiar with the pharmacokinetics and -dynamics of psychedelics. Can the authors perhaps explain their choice of variables for the dominance analysis.

Thank you for this suggestion. We have added the following to this section:

“While, as discussed above, the serotonin 2a receptor is the primary receptor responsible for much of psychedelics’ neural and subjective effects, their action at other serotonin receptors is of increasing interest to the field. Indeed rodent models have suggested that some of psychedelics’ antidepressant and synaptogenic effects may act through non-2a serotonergic mechanisms (Hesselgrave et al. 2021; Shao et al. 2021).”

29. Lines 396-399: It is not clear to me why the authors are referring to SI Figure 4 here? I think they mean SI Figure 5?

All references to SI figures have been updated as the numbering has changed.

30. Line 403: It would be great if the authors could include the literature references here.

We have added the following references at the requested location: (Li and Mashour 2019; Schartner et al. 2017; Timmermann et al. 2019; Eckernäs et al. 2023; Pallavicini et al. 2021)

31. Line 421: Can the authors add “...this is the third independent dataset including two previous published ones using LSD and psilocybin (Singleton et al. 2022) which has found...” to clarify this point?

We have made the requested change.

32. Lines 447-454: Can the authors include information on the participants in the supplementary materials including inclusion criteria. Specifically, it is important to understand if and why subjects were excluded regarding previous drug experiences.

Thank you for this suggestion, we now include the exclusion criteria in this section:

“Exclusion criteria included: <18 years old at the moment of participation, MR contraindications, absence of experience with a psychedelic, an adverse reaction to a psychedelic, history of psychiatric or physical illness rendering unsuitable for participation (i.e. diabetes, epilepsy, or heart disease), family history of psychotic disorder, or excessive use of alcohol or drugs of abuse.”

33. Methods section: The methods section is unfortunately not detailed enough to be able to completely comprehend what has been done without going through the provided code (and let's be honest here: which no reader does unless they are using it for their own study). Especially problematic is the fact that decisions are not justified including stating the lack of empirical evidence if applicable. I believe this is very important to make the field more transparent and guide decisions in future studies. In addition, the methods section is not well structured. The description of the methods is randomly distributed across the results section, the figures, the methods section, and references to previous studies. Thus, it becomes very difficult to gather all necessary information to understand what has been implemented and why. If there is not enough space, please move the methods section to the supplementary materials.

Thank you for this suggestion. We have re-organized our methods section to be more clear and to follow the order of the results section. We have also added more information throughout.

34. Lines 456-458: Please state how much time was between the first and the second session on each testing day.

Scanning sessions were separated at least 2 hours apart. The second scanning sessions occurred only after participants had fully returned to baseline. Furthermore, no participant underwent two consecutive DMT or placebo sessions on the same day. We have added this information to this methods section.

35. Line 462: Please add duration of administration. You can guess from Figure 1 but, unless I have not missed it, it is nowhere explicitly stated. Also, I think it should read "of the 8th minute".

We have updated this section to include the requested information.

"... injected over 30s, and then flushed with 10 mL of saline over 15s ..."

36. Lines 464-466: Can the authors please clarify how the rating was implemented exactly.

Specifically, the ratings were collected by playing an audio cue ("Intensity from 0 to 10") through MR-compatible earphones which was followed by the participants' verbal response collected via an MR-compatible microphone. We have updated the methods to reflect this.

37. Lines 466-467: Why did the authors choose a single-blind design? Please discuss and justify why blinding was chosen since one can criticize the futility of blinding subjects from their psychedelic experience due to ethical and pragmatical considerations as discussed here [Schenberg, E.E., Who is blind in psychedelic research? Letter to the editor regarding: blinding and expectancy confounds in psychedelic randomized controlled trials. Expert Rev Clin Pharmacol, 2021. 14(10): p. 1317-1319.]. Benefits potentially gained by blinding, though again,

unlikely in research with compounds that induce psychedelic states, could instead be acquired via assessment of expectations for the psychedelic experience and incorporated into analyses using tools validated for this purpose such as the Credibility/Expectancy Questionnaire [Devilly, G.J. and T.D. Borkovec, Psychometric properties of the credibility/expectancy questionnaire. J Behav Ther Exp Psychiatry, 2000. 31(2): p. 73-86.]. Have these assessments been considered?

A single-blind design was chosen to allow researchers to be aware of the condition and therefore be attentive to any situations requiring psychological assistance. As found in our previous work (<https://www.nature.com/articles/s41598-019-51974-4>), when participants were blinded cardiac activity in the placebo condition also increased as in the DMT condition at the moment of injection, thereby aiding in the control of arousal confounders. While expectancy was not acquired, it would be fruitful to do so in future research. We now include this information in the limitations section.

38. Lines 469-470: Here the authors should maybe include a discussion on the variability of the subjective intensity ratings across sessions to the other including the discussion of possible carry-over effects – see comment 14.

We now include this discussion here. Please see our response to comment 14.

39. Line 474: I generally do not find it appropriate if description of any part of the methods are provided only in a reference to which the reader may or may not have access. All relevant information should be provided in the paper at hand.

We now include the full EEG-fMRI acquisition details as presented in the original study.

40. Lines 490-509: Same criticism as above. This is just not enough information (even when considering the necessity to read additional two more papers while trying to figure out which information is specific to the data analyzed in the present study). I would like to see a justification for the specific preprocessing choices (e.g., why did the authors choose AFNI (3dvolreg) for motion correction? Why did the authors choose to register each volume to the most similar volume in a least square sense and not to the first?). It is extremely important that the field starts to become transparent.

We have updated this methods section with more details and justification.

“The same preprocessing pipeline as used in previous work with LSD and psilocybin (Singleton et al. 2022) and reported in Timmermann and colleagues (2023) was used here. Scrubbing and nuisance regression was used, which has been shown to be an effective denoising strategy in-part due to the removal of high-motion individuals (Parkes et al. 2018). Six out of 20 participants were discarded from group analyses due to excessive head movement during the 28 minute DMT scans (>20% of scrubbed volumes with a scrubbing threshold of frame-wise displacement (FD) of 0.4 (Power et al. 2014)), leaving 14 for analysis. Preprocessing steps consisted of 1) de-spiking (3dDespike, AFNI (Cox

1996)), 2) slice time correction (3dTshift, AFNI), 3) motion correction (3dvolreg, AFNI) by registering each volume to the most similar volume, in the least squares sense, to all others (minimizing the amount of transformation required for each volume), 4) brain extraction (BET, FSL (Smith et al. 2004)), 5) rigid body registration to anatomical scans, 6) non-linear registration to 2mm MNI brain (Symmetric Normalization (SyN), ANTS (Avants, Tustison, and Song 2009)), 7) scrubbing - using an FD threshold of 0.4 - with scrubbed volumes being replaced with the mean of the surrounding volumes. Additional preprocessing steps included: 8) spatial smoothing (FWHM) of 6mm (3dBlurInMask, AFNI), 9) band-pass filtering between 0.01 to 0.08 Hz (3dFourier, AFNI), 10) linear and quadratic de-trending (3dDetrend, AFNI), 11) regressing out 9 nuisance regressors (all nuisance regressors were bandpass filtered with the same filter as in step 9), out of these, 6 were motion-related (3 translations, 3 rotations) and 3 were anatomically-related. Specifically, the anatomical nuisance regressors were: a) ventricles (Freesurfer (Dale, Fischl, and Sereno 1999), eroded in 2 mm space), b) draining veins (FSL's CSF minus Freesurfer's ventricles, eroded in 1mm space (Smith et al. 2004; Dale, Fischl, and Sereno 1999)), and c) local white matter (WM) (FSL's WM minus Freesurfer's subcortical gray matter structures, eroded in 2mm space (Smith et al. 2004; Dale, Fischl, and Sereno 1999)). Lastly, global signal regression was performed and time-series were parcellated into 100 cortical (Schaefer et al. 2018) and 16 subcortical (Tian et al. 2020) regions of interest. Because global signal regression remains a hotly debated topic, particularly in the context of altered states of consciousness (Tanabe et al. 2020; Mortaheb et al. 2024; Singleton and Kuceyeski 2024), we repeated our main analyses without the use of global signal regression (SI Fig. 11)."

41. Lines 499-500: Why did the authors choose to use scrubbing and nuisance regression as a denoising strategy method? Please discuss (see e.g., Parkes et al., 2018, Neuroimage). How exactly did the authors regress out 9 nuisance regressors? What are the 3 anatomically-related regressors? How were they retrieved?

We have updated this methods section with more details and justification (see above).

42. Line 510: I am missing the details regarding EEG preprocessing. I am no expert on EEG but I am quite sure that you do need to preprocess the data before you can run Lempel Ziv.

We have added a section on EEG processing to the methods.

43. Line 522-525: Please discuss why the same parameters as in prior work were chosen.

This is to be consistent with our prior work on control energy in psychedelics. We have added this to the methods.

44. Lines 539-540: Again, I find it too much to ask for the reader to look up multiple different papers and figure out which information details fit the data reported in this manuscript. Please

provide this information in the present publication. If word restriction is an issue, put the methods into the supplementary materials.

We have updated this methods section.

45. Lines 544-551: Can the authors please add a short discussion why Lempel-Ziv was chosen as the measure of signal diversity compared to other approaches measuring entropy that have been introduced lately to detect the level of complexity in the brain as a guide for future implementation.

Lempel-Ziv complexity of EEG/MEG is a broadly replicated signature of psychedelic states ([10.1371/journal.pone.0242056](https://doi.org/10.1371/journal.pone.0242056), [10.1016/j.neuroimage.2019.03.076](https://doi.org/10.1016/j.neuroimage.2019.03.076), [10.1038/srep46421](https://doi.org/10.1038/srep46421)), including with DMT ([10.1038/s41598-019-51974-4](https://doi.org/10.1038/s41598-019-51974-4), [10.1002/psp4.12933](https://doi.org/10.1002/psp4.12933), [10.1177/0269881120981384](https://doi.org/10.1177/0269881120981384)). In addition, the parent study ([10.1073/pnas.2218949120](https://doi.org/10.1073/pnas.2218949120)) used this same method for their assessment of EEG signal complexity, and here we sought to understand how control energy relates to the previous findings rather than to introduce an entirely new assessment of EEG signal diversity in this dataset.

46. Line 552: Maybe it is better to put the section “Minimum control energy” here to follow the structure of the reported results and make it easier for the reader. Also add a section describing the stats for comparing global control energy between DMT and PCB as described in Figure 2a for which the authors present the results in the results section “Global control energy is lower after DMT infusion versus after placebo”. Then, add a section describing and briefly justifying the Pearson correlation implemented to obtain the results in section “Global control energy under DMT negatively correlates with subjective drug intensity ratings and signal diversity from simultaneous EEG” (it is not clear what the authors have done exactly – permutation?). It would be good if each section has its own header following the structure of the results – so the authors can refer to the specific method section when reporting the results.

We have systematically overhauled the organization of methods section to fit with the order of the results.

47. Lines 553-567: This part is very clear and well explained and also justified.

Thank you!

48. Lines 569-624: This part should move up to match the order in the results section (see comment above).

We have moved this section as suggested.

49. Lines 585-503: If I understand correctly, this part here concerns only the DMT simulation. To make this easier to relate to the results, I would move this part into its own section to match the structure of the results section. Also, this part is not 100% clear. As it is argued now, I do not

understand what kind of measures were exactly taking to “demonstrate the advantage of incorporating both the simulated brain-effect concentration (compared to simulated plasma concentration)” (lines 292-293). In Figure SI 5, results comparing these two approaches are shown. How was the simulated plasma concentration calculated and how does it differ from simulated brain-effect concentration (defined as “independently validated model of DMT’s pharmacokinetic impact on EEG alpha rhythms to simulate population-level ‘brain-effect’ of DMT over the course of the 28 minute scans (Eckernäs et al. 2023)”) to show this advantage?

We have moved this to its own section in the methods. We have added clarification to the methods:

“While plasma concentrations are based on observed/measured concentrations from collected blood samples, effect compartment concentrations are theoretical concentrations based on observed/measured effect data (in this case alpha waves). The effect compartment is a way to link plasma concentrations to effect when it is believed that the effects are driven by concentrations in the biophase rather than plasma itself. This can be the case when the time course of the effects are slightly delayed as compared to plasma concentrations, explained by the fact that it takes a certain time for the drug to reach the effect compartment (Sheiner et al. 1979). The regional serotonin 2a receptor density ρ was derived from previously published PET data (See *5-HT receptor mapping*) (Beliveau et al. 2017). In SI Fig 13, we repeated our simulation in two ways, using (a) DMT plasma concentration estimates as $DMT(t)$, and (b) using a vector for ρ that contains the same total sum as the one constructed of serotonin 2a receptor density, but uniformly spread across regions. By having better correspondence with the empirical DMT time-series, these alternative simulations demonstrate the advantages of using brain effect compartment concentrations over plasma concentrations, and skewing control by the serotonin 2a receptor distribution.”

50. Line 596: Should this not read SI Figure 5?

We have corrected this.

51. SI Figure 4: The @ as symbol for alpha is confusing.

Thank you. We have updated the figure to be more clear.

Reviewer #3 (Remarks to the Author):

This is a fascinating manuscript that would be a valuable contribution if published as is. However, there are a few issues that may be valuable to address.

Did the authors consider using the average, modal, and boundary controllability analyses described here? If not, why not? Perhaps a brief comment about further work in that direction could be helpful.

<https://www.nature.com/articles/ncomms9414/>

Thank you for your comments and this excellent suggestion. We did not consider these analyses here, as they would typically be applied in situations where the structural connectomes are different across individuals, however here we keep the structural connectome fixed across individuals (as individual participant diffusion data were not available) in order to analyze the influence of the different functional states visited under psychedelics and placebo from a network control perspective. It would be interesting to perhaps use either effective or functional connectivity as a connectome (instead of an average structural connectome) and to apply the suggested analyses. We have added this to the discussion on future work.

The description of dominance analyses was very interesting/compelling. Was this the first analysis technique used? If another method was used and found to be lacking, perhaps it could be included as a supplementary material? If this was the only technique used, a little more up front description of its a priori justification could be helpful for some potentially skeptical readers (although I am personally compelled).

Thank you for this suggestion. This was the first technique used. We now refer the reader to our discussion of dominance analysis in the materials and methods section where the following is discussed:

Dominance analysis provides two important advantages. The first is its multivariate nature: by considering all possible combinations of predictors together, it accounts for the possibility of synergistic interactions among them, such that more variance in the target is explained by considering them together, than by each in isolation. Conversely, if collinear predictors exist, DA will identify which of them provides the superior performance once they are considered together, and in the context of other predictors. In other words, DA allows predictors to contextualize each other when explaining the target, rather than treating each of them as existing in isolation. The second advantage is interpretability: the outcome measure of percentage relative importance allocates the variance explained to each predictor, enabling easy comparison between them (e.g., “predictor X explains twice as much variance as predictor Y”).

"In recent work using a receptor-informed network control theory framework, we demonstrated that the serotonergic psychedelics lysergic acid diethylamide (LSD) and psilocybin flatten the brain's control energy landscape in a manner that covaries with more dynamic and entropic brain activity."

In this previous work, more dynamic/entropic brain activity was interpreted as indicating a flattened energy landscape, in support of the REBUS model. However, an "ALBUS" model

provides an alternative (Entropic Brain Theory compatible) interpretation in which these more diverse states are reached via more powerful/persistent predictions, potentially especially those associated with intermediate-level conscious representations:

<https://psyarxiv.com/zqh4b/>

That is, rather than (or in addition to) flattening the terrain, psychedelics might also help with climbing peaks (and perhaps escaping from local optima). Do the methods used in this paper help to disambiguate these possibilities, or is this still an open question?

Thank you for this insightful comment. While our prior work does suggest that activity is more dynamic under psychedelics (for example - quicker state-transitions and less predictable sequences of states), our approach does not enable us to explicitly link this finding to a weakening (or strengthening in the case of ALBUS) of priors. Following peer-review of the aforementioned work, we addressed this ambiguity in our final discussion:

“As used here, the term “energy” denotes the magnitude of the input that needs to be injected into the system (the brain’s structural connectome) in order to obtain the desired state transition.... It is also not a direct measure of the brain’s variational free energy landscape, an information theoretic topic foundational to REBUS. We hypothesize that the empirical changes in control energy demonstrated here indicate a flattening of the posterior, which may or may not arise from a relaxation of prior beliefs and a flattening of the free energy landscape as posited by REBUS. Future work will be required to determine if any direct relationship exists between free energy and control energy.”

REVIEWERS' COMMENTS:

Reviewer #1 (Remarks to the Author):

My questions have been fully addressed by the authors. I believe the additional analyses and the more comprehensive and in-depth discussion have significantly improved the manuscript. I particularly appreciate that this EEG-fMRI dataset of DMT captures the entire administration process, allowing for a more comprehensive understanding of how DMT dynamically influences brain activity over time.

Although the revision process took a long time, I believe the wait was entirely worth it. I appreciate the authors' effort in refining and enhancing this work.

Minor Suggestions:

- The figure legends in the supplementary materials could be more detailed. For example, clarifying what analyses were performed in each figure, what specific indices represent, and the definitions of abbreviations would improve clarity.

Done!

- In Figures 7 and 8, some plots are not fully displayed, and certain data points appear to be cut off.

Done!

Reviewer #2 (Remarks to the Author):

The authors have answered all the questions and improved the manuscript fundamentally. With the additional information, there is one issue that still needs to be addressed:

1. Global control energy analyses: More information is needed for how the normalization procedures were implemented to calculate control energy. At least provide references.

Done!